# Selective Coupling of Decoupled Informative Regions: Masked Attention Alignment for Data-Free Quantization of Vision Transformers

Biao Qian [*1]   Yang Wang [*2]   Yong Wu [3]   Jungong Han [1]

## Abstract

Data-Free Quantization (DFQ) addresses data security concerns by synthesizing samples, without accessing real data. It has garnered increasing attention in the context of Vision Transformers (ViTs), owing to the superiority of the self-attention mechanism compared to classical convolutional operation. However, previous DFQ arts for ViTs often suffer from a *distribution mismatch* between synthetic samples and input distribution expected by quantized models $Q$, resulting in the suboptimal performance. In this paper, we propose a novel **Mask**ed Attention **A**lignment approach for Data-Free **Q**uantization of ViTs, named **MaskAQ**, revealing that: 1) the semantics in the self-attention mechanism is predominantly localized to a sparse subset of patches, called *informative regions*; 2) the informative regions dominate the mutual information between synthetic samples and $Q$'s outputs. To these ends, we incorporate differential entropy maximum over patch similarity of synthetic samples, to decouple informative regions from noisy background. To couple with varied $Q$, the informative regions are selected to align full-precision models with $Q$ via a masked attention alignment objective, thus yielding high-quality synthetic samples. Furthermore, a periodic sample refreshing strategy comes up to endow MaskAQ with the capacity to continually adapt to the evolving state of $Q$ throughout the training process, to preserve desirable mutual information with synthetic samples. Extensive experiments verify the merits of MaskAQ over state-of-the-art approaches across multiple backbones and downstream tasks. Our code is available at *https://github.com/hfutqian/MaskAQ*.

[*]Equal contribution [1]Department of Automation, Tsinghua University, China. [2]School of Computer Science and Information Engineering, Hefei University of Technology, China. [3]Li Auto, China. Correspondence to: Jungong Han <jghan@tsinghua.edu.cn>.

*Proceedings of the $43^{rd}$ International Conference on Machine Learning*, Seoul, South Korea. PMLR 306, 2026. Copyright 2026 by the author(s).

## 1. Introduction

Despite the remarkable performance of Vision Transformers (ViTs) (Dosovitskiy, 2020; Han et al., 2022; Khan et al., 2022) across various computer vision tasks (Zhang et al., 2021b; Ma et al., 2023; Wang et al., 2024; Qian et al., 2022; 2021), their deployment on resource-constrained edge devices is hindered by the high computational and memory costs of ViTs (Lin et al., 2021). Model quantization (Gholami et al., 2022; Liu et al., 2021c; Zhang et al., 2025) serves as a promising approach to reduce the model complexity, which converts the pre-trained full-precision model $P$ into the quantized model $Q$. Due to the quantization error, $Q$ derived from $P$ is expected to recover the performance via the calibration (Liu et al., 2021c) or fine-tuning operation (Hubara et al., 2018) upon the original training data. Unfortunately, in data-sensitive scenarios, such as social and medical domains, the original training data is not always accessible due to privacy and security issues (Liu et al., 2021a).

As a promising alternative, Data-Free Quantization (DFQ) (Xu et al., 2020; Choi et al., 2021; Zhong et al., 2022; Qian et al., 2023b;a; Dung et al., 2024; Kim et al., 2025) has been extensively explored to quantize $P$ without access to the original data. The success of existing DFQ arts for Convolutional Neural Networks (CNNs) critically benefits from the stable *distributional prior* from Batch Normalization (BN) statistics (Cai et al., 2020; Zhang et al., 2021a), to approximate the original data distribution for sample generation. Recently, ViT architectures have received the growing adoption, evidenced by their capacity for direct *global* dependency modeling through self-attention mechanism (Vaswani et al., 2017) over the locality of CNNs, which are well-positioned to break through the performance ceiling of current DFQ arts for practical application. Such fact substantially motivates a transition of DFQ from CNNs to ViTs (Li et al., 2022; 2023a; Ramachandran et al., 2024; Choi et al., 2025; Hu et al., 2024; Zhao et al., 2025; Zhong et al., 2025). These substantial efforts observed that the straightforward migration of DFQ from CNNs to ViTs suffers from a dramatic performance degradation, since ViTs, built on the Layer Normalization (LN) (Ba et al., 2016; Dosovitskiy, 2020), inherently lack the BN layer with distributional pri-

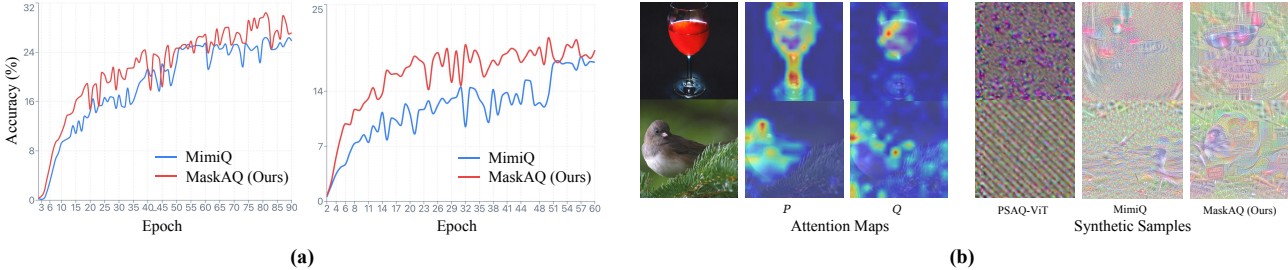

*Figure 1.* (a) Existing arts, *e.g.*, MimiQ (Choi et al., 2025), suffer from a obvious performance gap compared to our MaskAQ, owing to the suboptimal quality of synthetic samples. (b) The synthetic samples from existing arts, *e.g.*, PSAQ-ViT (Li et al., 2022) and MimiQ (Choi et al., 2025), usually exhibit two primary issues: *semantic dispersion*, where the semantics is distributed across the entire image, deviating from a coherent image structure; and *attentional disparity*, where $Q$ struggles to maintain attention alignment with $P$, owing to the absence of discriminative regions that are easily recognizable to $Q$ within synthetic samples. The experiments are conducted with DeiT-S and DeiT-T on the ImageNet dataset.

ors, resulting in low-quality synthetic samples. Meanwhile, the inherent sensitivity of the ViT architectures, particularly their reliance on precise inter-token relationships within self-attention mechanisms (Vaswani et al., 2017), further amplifies the quantization errors provided that the low-quality synthetic samples (*e.g.*, lack of semantic structures) generate ambiguous or noisy token representations.

To remedy these issues, the pioneer work, PSAQ-ViT (Li et al., 2022) and PSAQ-ViT V2 (Li et al., 2023a), deliver the first attempt to explore the prior information of the pre-trained ViTs to come up with a relative patch similarity metric to separate the foreground from the background for realistic samples. Following that, CLAMP-ViT (Ramachandran et al., 2024) further leverages a patch-level contrastive learning to capture meaningful inter-patch relationships. MimiQ (Choi et al., 2025) focuses on inter-head attention similarity to enhance the overall sample structure. Despite the significant strides of these approaches, a critical question remains unclear — *whether and how the synthetic samples for ViTs preserve crucial information for $Q$'s calibration*, potentially resulting in a pronounced performance disparity, especially for ultra-low precision, as illustrated in Fig.1(a). We identify two primary rationales (see Fig.1(b)) as follows: 1) *semantic dispersion*: the semantics of synthetic samples is distributed across the entire sample, deviating from a coherent image structure; 2) *attentional disparity*: the synthetic samples exhibit incomplete semantics, particularly the absence of discriminative regions that are easily recognizable to $Q$, such that $Q$ cannot focus on the semantically correct regions and struggle to align with $P$.

As motivated above, we propose a novel M̲asked Attention A̲lignment approach tailored for Data-Free Q̲uantization of ViTs, dubbed **MaskAQ**, as depicted in Fig.2, while *disclosing* that: 1) within the self-attention mechanism, the semantics is not uniformly distributed, but is highly concentrated within a sparse subset of image patches, termed

*Informative Region* (IR, Definition 1); 2) these informative regions serve as the *primary* contributors to preserve the desirable mutual information between synthetic samples and $Q$'s outputs[1], *i.e.*, enabling $Q$ to focus on informative regions. To this end, our basic idea is to couple the selected *informative regions* of synthetic samples with varying $Q$. Notably, MaskAQ features three primary components: *first*, we decouple informative regions from the noisy background within synthetic samples by maximizing the differential entropy of their patch similarity. *Second*, an adaptive masking mechanism for the informative regions comes up to couple with varying $Q$; upon such mask, we leverage a masked attention alignment objective to bridge the synthetic samples with $Q$, thereby facilitating the synthesis of high-quality samples. *Finally*, a periodic refreshing strategy comes up to maintain mutual information between synthetic samples and outputs for refined $Q$, enabling MaskAQ to continuously adapt to the evolution of $Q$ throughout the training process.

We theoretically and empirically demonstrate the effectiveness of MaskAQ and showcase the significant improvements over state-of-the-art approaches across multiple backbones and downstream tasks (*i.e.*, classification, detection and segmentation). Notably, under 3-bit case, MaskAQ achieves up to 3.1% Top-1 accuracy gains over existing arts on ImageNet for DeiT-T. The core contributions of this work are summarized as follows:

- We explicitly identify the gap or discrepancy between $P$ and $Q$, and observe two typical rationales, *i.e.*, semantic dispersion and attentional disparity, which pinpoints the critical bottleneck of prior arts.

- We propose MaskAQ, to revisit sample synthesis of

---

[1]Desirable mutual information commonly refers to maximizing the mutual information between synthetic samples and Q's outputs subject to an information budget constraint (see more details in Sec.2.4).

DFQ for ViTs from a novel information bottleneck (IB) perspective, which shifts the goal from approximating real data (prior DFQ work) to preserving the crucial information for calibrating Q.

- The IB analysis motivates several technical components, *i.e.*, informative region decoupling, masked attention coupling and periodic refreshing, enabling calibration-relevant sample synthesis.

- Extensive empirical evaluations across various backbones, downstream tasks and bit widths, demonstrate the superiority of our MaskAQ to existing DFQ approaches.

## 2. Masked Attention Alignment for Data-Free Quantization of ViTs

### 2.1. Preliminaries

To facilitate the understanding, we begin by outlining data-free quantization approaches for ViTs, mainly involving sample synthesis process.

#### 2.1.1. SAMPLE SYNTHESIS

DFQ approaches generally synthesize samples by exploiting the pre-trained full-precision models $P$. Given a synthetic sample $x \in \mathbb{R}^{H \times W}$ initialized from Gaussian noise, the one-hot ($OH$) loss is adopted to match the category information:

$$\mathcal{L}_{OH} = CE(z_p, y), \qquad (1)$$

where $z_p$ is the prediction output of $P$. $CE(\cdot, \cdot)$ represents the Cross-Entropy function and $y$ is the class label. To improve the image quality, the total variance (TV) loss (Rudin et al., 1992; Johnson et al., 2016) is adopted to promote enforce spatial smoothness as

$$\mathcal{L}_{TV} = ||x_{h,w} - x_{h+1,w}||_2^2 + ||x_{h,w} - x_{h,w+1}||_2^2, \quad (2)$$

where $h, w$ stands for the pixel position. Besides, the inter-head ($IH$) attention similarity loss (Choi et al., 2025) is utilized to enhance overall image structure during synthesis process, as formulated below:

$$\mathcal{L}_{IH} = \frac{1}{LN} \sum_{l=1}^{L} \sum_{n=1}^{N} (1 - \frac{1}{N_h^2} \sum_{i,j=1}^{N_h} D_{ssim}(A_{l,i,n}, A_{l,j,n})), \quad (3)$$

where $D_{ssim}(\cdot, \cdot)$ stands for the structural similarity index measure function (Wang et al., 2004). $L$, $N_h$ and $N$ are the total number of multi-head self-attention (MSA) layers, attention heads and image patches. Accordingly, $A_{l,i(j),n}$ denotes the attention map at the $l$-th MSA layer, $n$-th query position and $i(j)$-th attention head.

### 2.2. Masked Attention Alignment

#### 2.2.1. INFORMATIVE REGION DECOUPLING

As aforementioned, *semantic dispersion* is considered as a key challenge in data-free synthesis for ViTs, where gradients may spread over many patches, producing visually incoherent samples and diluting the supervision signal for calibrating $Q$. To address this, we introduce the following **Informative Region (IR)** as a principled, attention-driven decomposition of a synthetic sample into *high-value* (*i.e.*, semantically critical) patches and *noisy background* patches:

**Definition 1** (Informative Region). Given a synthetic sample $x$ with $N$ patches $\{x_n\}_{n=1}^{N}$ and their attention weights $\{\alpha_n\}_{n=1}^{N}$, we define the informative region as

$$IR = \{x_n \mid \alpha_n \geq \alpha_{[k_{ir}]}\}, \qquad (4)$$

where $\alpha_{[k_{ir}]}$ denotes the $k_{ir}$-th largest value among attention weights $\{\alpha_n\}_{n=1}^{N}$.

Notably, Definition 1 paves the cornerstone of our MaskAQ: it explicitly identifies where $P$ concentrates attention and enables us to separately regularize foreground structure and background noise during synthesis, which further endow us to decouple *informative* patches from background noise by encouraging *diverse and non-degenerate* attention patterns, inspired by (Li et al., 2022). Specifically, for layer $l$ of $P$, let $A_l^p \in \mathbb{R}^{N \times N}$ be the attention matrix. We compute pairwise cosine similarity between attention vectors:

$$S_{ij} = \frac{a_i \cdot a_j}{||a_i|| ||a_j||}, \quad i, j = 1, 2, \ldots, N, \qquad (5)$$

where $a_i$ is the $i$-th row of $A_l^p$, representing the attention distribution from patch $i$ to all patches. We then form a similarity distribution $p_l(s)$ by normalizing the histogram of $S_{ij}$ (excluding $i = j$), and compute the Shannon entropy:

$$H(p_l) = -\sum_k p_l(s_k) \log p_l(s_k), \qquad (6)$$

where $s_k$ are the discrete values of similarity. By maximizing $H(p_l)$, it avoids collapsed attention similarities and promotes diverse attention vectors, which empirically leads to clearer separation between informative patches and noisy background. We thus define the similarity-entropy regularizer as

$$\mathcal{L}_{fb} = -\sum_{l=1}^{L} H(p_l). \qquad (7)$$

Since the histogram estimator in Eq.(7) may introduce binning sensitivity and less stable gradients during synthesis, we adopt the Gaussian differential entropy for optimization stability. Following an information-theoretic approximation, the similarity distribution between attention vectors is approximately Gaussian, *i.e.*, $\mathcal{N}(\mu_l, \sigma_l^2)$. In practice, $\sigma_l^2$

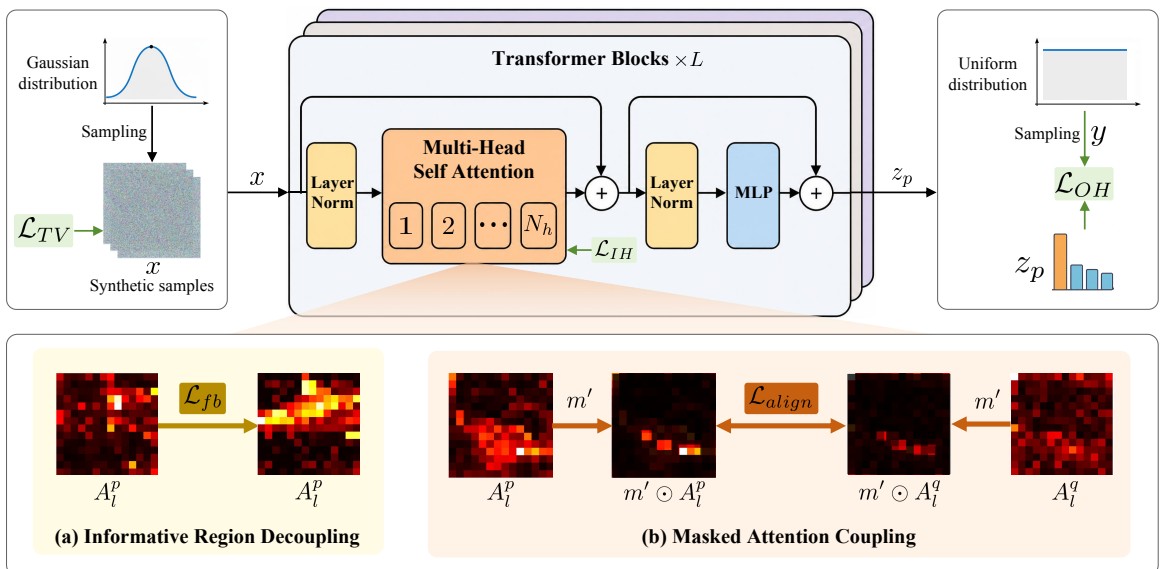

*Figure 2.* Illustration of our proposed MaskAQ, where our basic idea is to couple the selected informative regions of synthetic samples $x$ with varying $Q$, yielding high-quality of synthetic samples. MaskAQ is achieved via two primary components: (a) *informative region decoupling* (Sec.2.2.1) to construct coherent image structure via differential entropy maximum ($\mathcal{L}_{fb}$ in Eq.(10)) over patch similarity; and (b) *masked attention coupling* (Sec.2.2.2) to leverage a masked attention alignment objective equipped with an adaptive mask $m'$ ($\mathcal{L}_{align}$ in Eq.(14)), which aligns attention maps ($A_l^p$ and $A_l^q$) from $l$-th layer of $P$ and $Q$, thus bridging the synthetic samples with the varying state of $Q$.

is computed as the empirical variance of all off-diagonal pairwise cosine similarities $S_{ij}$, namely

$$\sigma_l^2 = \frac{1}{M} \sum_{i \neq j} (S_{ij} - \mu_l)^2, \tag{8}$$

$$\mu_l = \frac{1}{M} \sum_{i \neq j} S_{ij}, M = N(N-1). \tag{9}$$

Then its differential entropy surrogate is $H_l = \frac{1}{2} \log(2\pi e\, \sigma_l^2)$ (Cover, 1999; Shannon, 1948), where $2\pi e$ is a constant. Therefore, Eq. (7) can be equivalently formulated as

$$\mathcal{L}_{fb} = -\frac{1}{L} \sum_{l=1}^{L} H_l. \tag{10}$$

Eq. (10) serves as a smooth and tractable surrogate for the similarity-entropy regularizer in Eq.(7). Minimizing Eq. (10) regularizes the distributional geometry of attention vectors (Tishby et al., 2000), which strengthens the structural coherence of synthesized samples and makes the informative region more distinguishable; see Theorem 2 and Fig. 2(a).

### 2.2.2. MASKED ATTENTION COUPLING

The informative region (IR) identified by $P$ serves as a reliable semantic anchor for sample synthesis. However, under ultra-low precision, *attentional disparity* is regarded as another key challenge, where the quantization operation may

distort the self-attention dynamics of $Q$, such that directly aligning full attention maps tends to over-regularize irrelevant background patches and may drive synthesized samples away from real distribution. Based on IR, we propose a masked attention alignment objective to couple synthesis process with the *current state* of $Q$. Specifically, we construct an *adaptive mask* to select a subset of informative patches and enforce the desirable mutual information between $P$ and $Q$ within these selected regions through *an alignment objective*.

**Adaptive mask generation**. According to **IR** (**Definition 1**), the regions attended by $P$ and $Q$ differ substantially, especially under low precision; moreover, due to quantization-induced perturbations, $Q$ may fail to consistently focus on the target regions. Therefore, we construct a binary mask $m \in \{0, 1\}^N$ from $P$'s attention weights as

$$m[n] = \begin{cases} 1 & \alpha_n \geq \alpha_{[k]}, \\ 0 & otherwise, \end{cases} \tag{11}$$

where $m[n]$ denotes $n$-th element of $m$. $\alpha_{[k]}$ represents the $k$-th largest value in the attention weights $\{\alpha_1, \alpha_2, \ldots, \alpha_N\}$ corresponding to the $N$ patches of $x$. Notably, $m$ selects a *stricter* subset of informative patches from Definition 1 to prevent over-regularization, hence $\alpha_{[k]} > \alpha_{[k_{ir}]}$. To avoid overfitting to a fixed set of attended patches, we further incorporate a *stochastic dropping* strategy to update the mask during synthesis. Given $m$ of each sample, we collect

selected indices

$$\mathcal{P} = \{n \in \{1, 2, \ldots, N\} \mid m[n] = 1\}. \quad (12)$$

Based on a dropout probability $p_{\text{drop}} \in [0, 1)$ that introduces a moderate level of stochasticity, the number of positions to retain is determined as

$$k_{\text{keep}} = \max\left(k_{\min}, \lfloor |\mathcal{P}| \cdot (1 - p_{\text{drop}}) \rfloor\right), \quad (13)$$

where $k_{\min}$ is a predefined minimum, serving as a safeguard to avoid dropping all patches; $\lfloor \cdot \rfloor$ is the floor function. Then, we randomly select $k_{\text{keep}}$ retention positions from $\mathcal{P}$ without replacement and denote their index as $\mathcal{P}_{\text{keep}}$, such that $m$ is updated to $m'$.

**Alignment objective**. The masked regions indicated by $m'$ provide the most relevant spatial support for interacting with $Q$. We enforce consistency between $P$ and $Q$ within these regions by minimizing

$$\mathcal{L}_{align} = \frac{\sum_{l=1}^{L} \|m' \odot (A_l^p - A_l^q)\|_1}{\|m'\|_0}, \quad (14)$$

where $A_l^p$ and $A_l^q$ are the attention maps of $l$-th layer in $P$ and $Q$. $\|\cdot\|_1$ and $\|\cdot\|_0$ are $\ell_1$ and $\ell_0$ norm; $\odot$ denotes element-wise multiplication. By minimizing Eq.(14), the masked regions within synthetic samples are optimized to capture the semantics relevant to $Q$; see Theorem 2 and Fig.2(b).

### 2.3. Overall Pipeline

#### 2.3.1. INFORMATIVE REGION ORIENTED SYNTHESIS

Taking Eq.(10) and Eq.(14) all together, the overall optimization objective for sample synthesis is confirmed as

$$\mathcal{L}_S = \mathcal{L}_{prior} + \lambda_{fb}\mathcal{L}_{fb} + \lambda_{align}\mathcal{L}_{align}, \quad (15)$$

where $\lambda_{fb}$ and $\lambda_{align}$ are the balancing parameters. $\mathcal{L}_{prior}$ is defined as the combination of Eq.(1), Eq.(2) and Eq.(3), which is further utilized to incorporate prior information into the synthetic samples.

We remark that the attention alignment over *masked informative regions* improves the structural coherence and reduces distributional discrepancies between synthetic samples and the expected input of varying $Q$. Subsequently, $Q$ is optimized through masked calibration on these samples to restore its performance, as discussed in the next.

#### 2.3.2. MASKED CALIBRATION

Following the above, considering the distinct response outputs of both $P$ and $Q$ to the informative versus non-informative regions (Definition 1), we prioritize the former during calibration process. To be analogous to Eq.(11),

---

**Algorithm 1 MaskAQ: Masked Attention Alignment for Data-Free Quantization of ViTs**

**Require:** Initial synthetic samples $x$; $P$ and $Q$ parameterized by $\theta_p$ and $\theta_q$; an initial learning rate $\eta$; an initial binary mask $m'$.
1: **for** Refreshing Number **do**
2:    **for** Iteration Number of Synthesis **do**
3:       Compute adaptive binary mask $m'$
4:       Estimate sample synthesis loss $\mathcal{L}_S$ via Eq.(15)
5:       Update synthetic samples $x$
6:    **end for**
7:    Obtain the synthesized samples $x$
8:    **for** Iteration Number of Calibration **do**
9:       Estimate calibration loss $\mathcal{L}_Q$ via Eq.(16)
10:      Update $Q$: $\theta_q \leftarrow \theta_q - \eta \frac{\partial \mathcal{L}_Q}{\partial \theta_q}$
11:      Update learning rate $\eta$
12:    **end for**
13: **end for**
**Ensure:** Quantized model $Q$.

---

we can obtain the binary mask $m_l^c$ of $l$-th layer for calibration process, such that the patch weight factor $w_l = 1 + m_l^c \cdot (w - 1)$, where $w$ is the weight assigned to the informative regions. Finally, $Q$ is calibrated via the following optimization objective:

$$\mathcal{L}_Q = \frac{1}{L\,N_h} \sum_{l=1}^{L} \sum_{n_h=1}^{N_h} \frac{\sum_{n=1}^{N} w_{l,n}\, D\left(h_{l,n_h,n}^p,\, h_{l,n_h,n}^q\right)}{\sum_{n=1}^{N} w_{l,n}}, \quad (16)$$

where $h_{l,n_h}^p$ and $h_{l,n_h}^q$ are $n_h$-th attention head outputs of the $l$-th layer in $P$ and $Q$. $D(\cdot, \cdot)$ represents the similarity metric function.

#### 2.3.3. PERIODIC SAMPLE REFRESHING

It is worth noting that, as $Q$ is continually optimized to recover the performance during calibration process (Eq.(16)), the initial synthetic samples may become ineffective for $Q$'s current state. To preserve desirable mutual information between synthetic samples and updating $Q$, we develop a periodic refreshing strategy, which endows MaskAQ with the capacity to continually adapt to the evolving state of $Q$. Specifically, we regularly update the synthetic samples via Eq.(15) at fixed intervals throughout the training process. The whole training process is summarized in Algorithm 1.

### 2.4. Theoretical Analysis: Information Bottleneck Perspective

One may wonder why MaskAQ benefits $Q$'s calibration. We provide further insights into MaskAQ from information bottleneck (IB) perspective (Tishby et al., 2000; Alemi

et al., 2017). Formally, given synthetic samples $x$, the output representations of $P$ and $Q$ are denoted as $z_p$ and $z_q$. According to the information bottleneck principle, an ideal quantization aims to *compress* the input information from $x$ while *preserving* predictive information, thus the mutual information for $P$ and $Q$ meets the followings:

$$I(x; z_q) \leq I(x; z_p) \text{ and } I(z_q; y) \approx I(z_p; y), \quad (17)$$

where $I(\cdot; \cdot)$ is utilized to calculate the mutual information and $y$ is the task target. To make the approximation in Eq.(17) precise, we view quantization as imposing an information budget. Let $C$ denote an upper bound governed by the quantization precision (*e.g.*, bit width), such that $I(x; z_q) \leq C$. Then the optimal quantization process is to solve the following objective

$$\max I(z_q; y), \quad \text{s.t. } I(x; z_q) \leq C. \quad (18)$$

Eq.(18) suggests that the limited representational ability should be allocated to the most informative regions as per Definition 1, thereby requiring $Q$ to focus on those regions in synthetic samples. To understand how $Q$ retains predictive information of $P$, we offer the following theorem:

**Theorem 1** (Informative Region Alignment). *Let $a_r$ denote the informative-region variable of real samples, and let $z_p^r$ and $z_q^r$ be the corresponding prediction outputs of $P$ and $Q$, respectively. Assume that for some $\varepsilon_r \leq \frac{1}{2}$,*

$$D_{\text{TV}}\Big(p_{\theta_q}(z_q^r, y \mid a_r),\ p_{\theta_p}(z_p^r, y \mid a_r)\Big) \leq \varepsilon_r, \quad (19)$$

$$D_{\text{TV}}\Big(p_{\theta_q}(a_r, y \mid z_q^r),\ p_{\theta_p}(a_r, y \mid z_p^r)\Big) \leq \varepsilon_r, \quad (20)$$

*where $D_{\text{TV}}(\cdot, \cdot)$ denotes the total variation (TV) distance. Then $Q$ approximately preserves $P$'s predictive information on informative regions in the sense that*

$$\big|I_{\theta_q}(z_q^r; y) - I_{\theta_p}(z_p^r; y)\big| \ \leq \ \Delta_r(\varepsilon_r), \quad (21)$$

*where $\Delta_r(\varepsilon_r)$ vanishes as $\varepsilon_r \to 0$.*

The proof is deferred to Appendix B.2. Theorem 1 suggests that the alignment between $P$ and $Q$ on informative regions of real samples is sufficient to maintain predictive power. However, under data-free scenarios, we have access to synthetic samples rather than real ones. This naturally leads to a critical question: under what conditions can the synthetic samples serve as effective substitutions for real samples in preserving mutual information with respect to $y$? The following theorem claims the case of synthetic samples upon the informative regions.

**Theorem 2** (Synthetic Sample upon Informative Regions). *Let $a_s$ denote the informative-region variable of synthetic samples $x_s$, and let $z_q^s$ be the corresponding prediction outputs of $Q$. If there exists $\varepsilon_s \leq \frac{1}{2}$ such that*

$$D_{\text{TV}}\Big(p_{\theta_q}(z_q^s, y \mid a_s),\ p_{\theta_p}(z_p^r, y \mid a_r)\Big) \leq \varepsilon_s, \quad (22)$$

$$D_{\text{TV}}\Big(p_{\theta_q}(a_s, y \mid z_q^s),\ p_{\theta_p}(a_r, y \mid z_p^r)\Big) \leq \varepsilon_s, \quad (23)$$

*and the region-label informativeness between synthetic and real informative regions is matched:*

$$\big|I(a_s; y) - I(a_r; y)\big| \leq \xi, \quad (24)$$

*then synthetic samples enable $Q$ to approximate $P$ in predictive information:*

$$\big|I_{\theta_q}(z_q^s; y) - I_{\theta_p}(z_p^r; y)\big| \ \leq \ \Delta_s(\varepsilon_s) + \xi + \Delta_a(\varepsilon_s), \quad (25)$$

*where $\Delta_s(\varepsilon_s)$ and $\Delta_a(\varepsilon_s)$ vanish as $\varepsilon_s \to 0$.*

The proof is deferred to Appendix B.3. Theorem 2 implies that the synthetic samples are required to produce meaningful semantics within their informative regions and promote the attention alignment between $P$ and $Q$.

We *claim* the above analysis to be consistent with the insights on $\mathcal{L}_{fb}$ of Eq.(10) and $\mathcal{L}_{align}$ of Eq.(14): *first*, $\mathcal{L}_{fb}$ promotes semantic aggregation towards *informative regions* within synthetic samples, capitalizing on the inherent sparsity of the self-attention mechanism, which is designed to satisfy the condition $|I(a_s; y) - I(a_r; y)| \leq \xi$ in Eq.(24). *Second*, anchored on *masked informative regions*, $\mathcal{L}_{align}$ enforces the attention alignment between $P$ and $Q$, helping satisfy the conditions in Eq.(22) and Eq.(23). Therefore, we can obtain desirable synthetic samples via Eq.(15) to benefit $Q$'s calibration.

## 3. Experiment

### 3.1. Experimental Settings and Details

To evaluate MaskAQ, we conduct the experiments on the typical image classification dataset, *i.e.*, ImageNet (ILSVRC2012) (Russakovsky et al., 2015), which contains 1.2M training images and 50k validation images from 1000 categories; while the results on object detection and semantic segmentation tasks are provided in Appendix C.1. Notably, merely validation sets are used to evaluate quantized models $Q$ under data-free setting. We adopt the different variants (*e.g.*, the tiny, small, and base versions) of ViT (Dosovitskiy, 2020), DeiT (Touvron et al., 2021), and Swin Transformer (Liu et al., 2021b) as backbone networks.

**Quantization**. Following (Choi et al., 2025), we employ a uniform quantization strategy, which converts floating-point values $\theta_p$ of both weights and activations from full-precision models $P$ into integers $\theta_q$ for quantized models $Q$ below:

$$\theta_q = clip(\lfloor \theta_p \cdot s - z \rceil, T_{min}, T_{max}), \quad (26)$$

where $s$ and $z$ is the scale factor and the zero point. $\lfloor \cdot \rceil$ is the round operator; $clip(\cdot, \cdot, \cdot)$ denotes the clip operation with minimum $T_{min}$ and maximum $T_{max}$ of the clip range.

*Table 1.* Classification accuracy (%) comparison with state-of-the-art approaches on the ImageNet dataset. †: the results reproduced by author-provided code; -: no results are reported. FP: full precision. $m$w$n$a indicates the weights and activations are quantized to $m$-bit and $n$-bit. The best results are reported with **boldface**.

| Bit width | Methods | Venue | Target Architecture | Backbone Networks | | | | | |
|---|---|---|---|---|---|---|---|---|---|
| | | | | ViT-T | ViT-B | DeiT-T | DeiT-S | DeiT-B | Swin-T |
| FP | Baseline | - | - | 72.01 | 84.53 | 72.21 | 79.85 | 81.85 | 81.35 |
| *3w3a* | MimiQ (Choi et al., 2025) | AAAI' 25 | ViT | 8.64† | 41.28† | 19.55† | 27.39† | 41.86† | 42.90† |
| | **MaskAQ (Ours)** | - | ViT | **11.50** | **43.39** | **22.65** | **30.41** | **43.28** | **44.98** |
| *4w4a* | GDFQ (Xu et al., 2020) | ECCV' 20 | CNN | 2.95 | 11.73 | 25.96 | 22.12 | 30.04 | 42.08 |
| | Qimera (Choi et al., 2021) | NeurIPS' 21 | CNN | 0.57 | 5.61 | 15.18 | 11.37 | 32.49 | 47.98 |
| | AdaDFQ (Qian et al., 2023a) | CVPR' 23 | CNN | 2.00 | 6.21 | 19.57 | 14.44 | 19.22 | 38.88 |
| | PSAQ-ViT (Li et al., 2022) | ECCV' 22 | ViT | 0.67 | 0.94 | 19.61 | 5.90 | 8.74 | 22.71 |
| | PSAQ-ViT V2 (Li et al., 2023a) | T-NNLS' 23 | ViT | 1.54 | 2.83 | 22.82 | 32.57 | 45.81 | 50.42 |
| | GenQ (Li et al., 2024) | ECCV' 24 | ViT | - | 67.50 | - | - | **76.10** | - |
| | CLAMP-ViT (Ramachandran et al., 2024) | ECCV' 24 | ViT | - | - | 0.15 | 0.13 | - | - |
| | MimiQ (Choi et al., 2025) | AAAI' 25 | ViT | 42.99 | 62.91 | 52.03 | 62.72 | 74.10 | 69.33 |
| | **MaskAQ (Ours)** | - | ViT | **44.90** | **70.53** | **53.10** | **63.81** | 74.41 | **70.40** |
| *5w5a* | GDFQ (Xu et al., 2020) | ECCV' 20 | CNN | 24.40 | 33.56 | 44.76 | 57.00 | 71.03 | 61.30 |
| | Qimera (Choi et al., 2021) | NeurIPS' 21 | CNN | 26.70 | 9.43 | 33.13 | 33.65 | 47.01 | 62.13 |
| | AdaDFQ (Qian et al., 2023a) | CVPR' 23 | CNN | 27.10 | 43.02 | 53.85 | 59.55 | 71.12 | 64.61 |
| | PSAQ-ViT (Li et al., 2022) | ECCV' 22 | ViT | 17.66 | 16.80 | 53.36 | 47.35 | 57.23 | 58.63 |
| | PSAQ-ViT V2 (Li et al., 2023a) | T-NNLS' 23 | ViT | 40.21 | 74.29 | 55.18 | 65.30 | 73.16 | 69.77 |
| | CLAMP-ViT (Ramachandran et al., 2024) | ECCV' 24 | ViT | - | - | 5.42 | 3.19 | - | - |
| | MimiQ (Choi et al., 2025) | AAAI' 25 | ViT | 62.40 | 78.09 | 63.40 | 72.59 | 78.20 | 76.39 |
| | **MaskAQ (Ours)** | - | ViT | **63.01** | **78.60** | **64.27** | **73.18** | **78.89** | **76.82** |
| *4w8a* | GDFQ (Xu et al., 2020) | ECCV' 20 | CNN | 62.65 | 81.68 | 65.82 | 76.49 | 80.03 | 78.90 |
| | Qimera (Choi et al., 2021) | NeurIPS' 21 | CNN | 61.80 | 63.22 | 61.90 | 70.10 | 72.38 | 73.93 |
| | AdaDFQ (Qian et al., 2023a) | CVPR' 23 | CNN | 64.67 | 82.43 | 67.71 | 76.92 | 80.49 | 79.70 |
| | PSAQ-ViT (Li et al., 2022) | ECCV' 22 | ViT | 59.59 | 67.74 | 66.16 | 76.56 | 80.05 | 79.06 |
| | PSAQ-ViT V2 (Li et al., 2023a) | T-NNLS' 23 | ViT | 66.78 | 84.02 | 68.23 | 78.27 | 81.15 | 79.98 |
| | CLAMP-ViT (Ramachandran et al., 2024) | ECCV' 24 | ViT | - | - | 65.71 | 76.44 | - | - |
| | MimiQ (Choi et al., 2025) | AAAI' 25 | ViT | 68.15 | 84.20 | 69.86 | 78.48 | 81.34 | 80.06 |
| | **MaskAQ (Ours)** | - | ViT | **68.36** | **84.30** | **69.95** | **79.44** | **81.40** | **80.19** |

**Implementation Details**. For *sample synthesis*, the Adam optimizer is adopted with a learning rate of 0.1, momentum coefficients of $\beta_1$=0.9 and $\beta_2$=0.999, and a batch size of 32. We produce 10,000 synthetic samples and optimize each batch for 2,000 steps with $\alpha$=1.0 and $\beta$=2.5e-5. For *network calibration*, the SGD optimizer is employed with Nesterove momentum of 0.9, a learning rate of 1e-3 and a batch size of 16; $Q$ is calibrated for 200 epochs in total. Besides, for both sample synthesis and network calibration processes, we employ data augmentations from SimCLR (Chen et al., 2020). Regarding hyper-parameters, $\lambda_{fb}$ and $\lambda_{align}$ in Eq.(15), the weight $w$, are set as 1.0, 0.1 and 2, respectively. In particular, we dynamically anneal $k$ in Eq.(11) over training, reducing it from 50% to 10% of $N$, thus facilitating the attention alignment. $k_{\min}$ and $p_{\text{drop}}$ in Eq.(13) are set as 1 and 0.3. All experiments are implemented with the pytorch framework (Paszke et al., 2019) based on the code of (Choi et al., 2025), and run on an NVIDIA A800 GPU and an Intel(R) Xeon(R) Gold 6342 CPU @ 2.80GHz. ***Due to page limitations, more discussions and results are accessible in Appendix C.***

## 3.2. Comparison with the State-of-the-arts

To verify the superiority of our MaskAQ, we compare it with the typical DFQ arts, including: 1) GDFQ (Xu et al., 2020), Qimera (Choi et al., 2021) and AdaDFQ (Qian et al., 2023a) are targeted at traditional Convolutional Neural Networks (CNNs); 2) PSAQ-ViT (Li et al., 2022) exploits the patch similarity to generate realistic samples; upon that, PSAQ-ViT V2 (Li et al., 2023a) improve the sample diversity via adversarial training; CLAMP-ViT (Ramachandran et al., 2024) further considers image-patch-level contrastive learning to synthesize samples; 3) GenQ (Li et al., 2024) employs the advanced generative models to obtain the synthetic samples; 4) MimiQ (Choi et al., 2025) focuses on inter-head attention similarity for sample synthesis. In particular, GenQ only reports the results under 4w4a in the original paper. Besides, we adopt the results of GDFQ, Qimera, AdaDFQ and CLAMP-ViT for ViTs reported in MimiQ. Notably, we reproduce the results of MimiQ under the 3-bit setting using their official code, while most other approaches become highly unstable at 3w3a and tend to collapse to near-random accuracy, hence their results are not

reported. Since the retained information is more severely compressed, and the distribution mismatch between synthetic samples and inputs expected by $Q$ becomes larger, 3-bit cases exhibit the marked performance gap compared to other bit widths, hindering practical application. Nonetheless, we present their experimental results to highlight the efficacy of our MaskAQ, *i.e.*, preserving crucial information for $Q$'s calibration.

Tab.1 summarizes our findings as follows: *first*, MaskAQ delivers consistent and substantial accuracy gains over state-of-the-art baselines, verifying the effectiveness of coupling the selected informative regions to yield high-quality synthetic samples. Impressively, MaskAQ exhibits at most 3.10% accuracy gains on ImageNet compared to the competitors. GDFQ, Qimera and AdaDFQ designed for CNNs suffer from the poor performance or convergence when applied to ViT architectures, implying the necessity of MaskAQ to develop customized techniques. Besides, MaskAQ receives larger accuracy improvements (at least 18.91% with 4w4a) over PSAQ-ViT and PSAQ-ViT V2, since MaskAQ is capable of mitigating the semantic dispersion within synthetic samples. Although GenQ introduces extra generative models to yield abundant synthetic data, MaskAQ still achieves comparable performance with GenQ, highlighting the effectiveness of improving the quality of synthetic samples. Under the same settings, CLAMP-ViT holds an obvious performance margin (at least 3%) with MaskAQ. In particular, MaskAQ outperforms MimiQ with the advantages of at most 7.62%, benefiting from its focus on the attentional disparity between $P$ and $Q$ (Sec.2.2.2). *Second*, MaskAQ holds continuous accuracy improvements over other approaches across varied backbone networks, which is consistent with our proposal of regularly refreshing the synthetic samples to adapt to the evolving state of $Q$ throughout the training process (Sec.2.3.3). As expected, MaskAQ maintains the similar accuracy gains (at least 1.42%) with varied network architectures, *e.g.*, ViT, DeiT, and Swin Transformer. *Finally*, MaskAQ delivers the pronounced improvements under ultra-low bit widths, *e.g.*, 3 bits, confirming the *necessity* of maintaining the sufficient mutual information between synthetic samples and varying $Q$ (Sec.2.4). Notably, for 3w3a, MaskAQ achieves substantial accuracy advantages (at most 3.1%) with DeiT-T compared to MimiQ, which, in turn, reflects the non-negligible attentional disparity between $P$ and varying $Q$, especially for low-bit scenarios, in line with our analysis on the limitations of synthetic samples in Sec.1.

### 3.3. Ablation Studies

#### 3.3.1. DISCUSSION ON EACH COMPONENT OF MASKAQ

We further validate the effectiveness of several components constituting MaskAQ from the following aspects: *w/o $\mathcal{L}_{fb}$*

*Table 2.* Ablation study for classification accuracy (%) about several components of MaskAQ. *m*w*n*a indicates the weights and activations are quantized to *m*-bit and *n*-bit. The best results are reported with **boldface**.

| $\mathcal{L}_{fb}$ | $\mathcal{L}_{align}$ | Sample Refreshing | $\mathcal{L}_Q$ | ViT-T (**4w4a**) | DeiT-T (**3w3a**) |
|---|---|---|---|---|---|
| ✗ | ✓ | ✓ | ✓ | 43.62 | 20.97 |
| ✓ | ✗ | ✓ | ✓ | 43.33 | 20.60 |
| ✓ | ✓ | ✗ | ✓ | 44.11 | 21.43 |
| ✓ | ✓ | ✓ | ✗ | 44.45 | 21.72 |
| ✓ | ✓ | ✓ | ✓ | **44.90** | **22.65** |

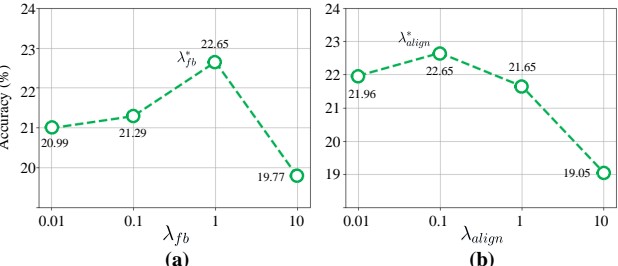

*Figure 3.* Ablation study for classification accuracy (%) about the hyper-parameters (a) $\lambda_{fb}$ and (b) $\lambda_{align}$.

in Eq.(10); *w/o $\mathcal{L}_{align}$* in Eq.(14); $\mathcal{L}_Q$ *w/o* binary mask in Eq.(16); abandoning the periodic sample refreshing strategy during the training process; and our MaskAQ. The ablation experiments are conducted with ViT-T and DeiT-T on ImageNet. Tab.2 suggests the obvious accuracy advantages (at least 0.45% and 0.93%) of MaskAQ over other cases. Notably, the case *w/o $\mathcal{L}_{align}$* suffers from the largest performance degradation (1.57% and 2.05%) compared to MaskAQ, implying that the distribution alignment between synthetic samples and $Q$'s input distribution play a critical role in synthesizing high-quality samples (Sec.2.2.2). Meanwhile, the case *w/o $\mathcal{L}_{fb}$* damages the model performance (*e.g.*, 22.65% to 20.97%), confirming the *importance* of the coherent image structure with synthetic samples. Regarding the calibration process, removing the mask in $\mathcal{L}_Q$ receives a relative small accuracy loss, indicating the priority of the informative regions when optimizing $Q$ (Sec.2.3.2). As a byproduct, abandoning the sample refreshing strategy leads to 0.79% and 1.22% accuracy loss, since $Q$ is dynamically updated during the calibration process, in line with Sec.2.3.3.

#### 3.3.2. PARAMETER STUDIES

We further investigate the effect of the parameters $\lambda_{fb}$ and $\lambda_{align}$ in Eq.(15), which are utilized to balance the importance of $\mathcal{L}_{fb}$ and $\mathcal{L}_{align}$ during the sample synthesis process. To evaluate the parameter impacts, we perform the ablation experiments based on DeiT-T. Specifically, we set varied

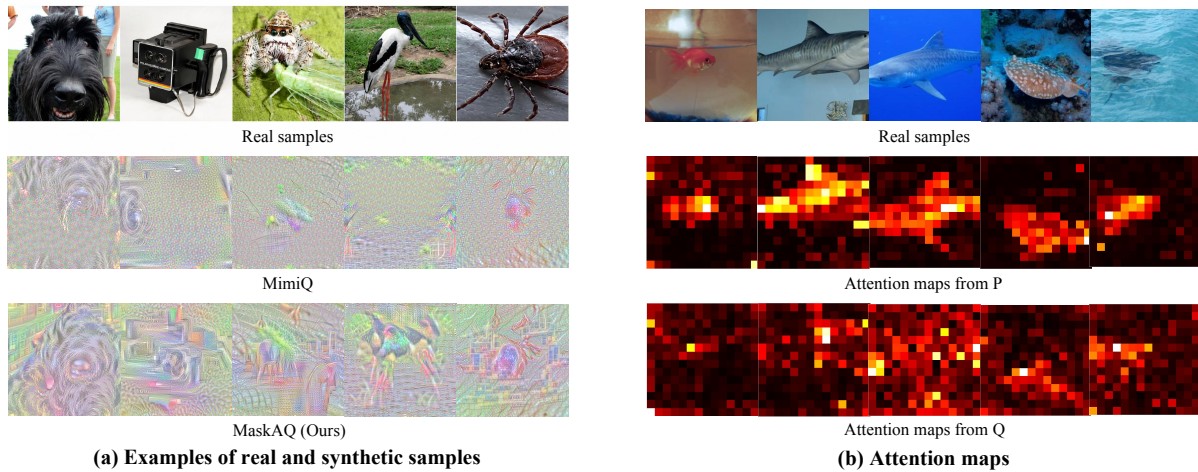

(a) Examples of real and synthetic samples    (b) Attention maps

*Figure 4.* (a) Examples of synthetic samples from MimiQ (Choi et al., 2025) and our MaskAQ, apart from the corresponding real samples. (b) Visualization of attention maps (the brighter, the larger) from the intermediate layer of $P$ and $Q$ based on MaskAQ.

$\lambda_{fb} \in \{0.1, 1, 10\}$ and $\lambda_{align} = \{0.01, 0.1, 1, 10\}$, thereby exploring a comprehensive grid of hyperparameter configurations. Fig.3 illustrates that the optimal performance is achieved with $\lambda^*_{fb} = 1$ and $\lambda^*_{align} = 0.1$, implying that $\mathcal{L}_{fb}$ for informative region decoupling and $\mathcal{L}_{align}$ for masked attention coupling can effectively mitigate the semantic dispersion and attentional disparity during the sample synthesis, enabling high-quality synthetic samples. The above fact confirms the necessity of *masked attention alignment between P and Q in Sec.2.2.1 and Sec.2.2.2*.

### 3.4. Insights into MaskAQ from Visualization

To provide further insights into MaskAQ, we perform the visual analysis on synthetic samples and attention maps with DeiT-T on ImageNet. Fig.4(a) illustrates the examples of synthetic samples from MimiQ (Choi et al., 2025) and our MaskAQ, apart from the corresponding real samples; while Fig.4(b) visualizes the attention maps from the intermediate layer of $P$ and $Q$. It is observed that, compared to existing arts, *e.g.*, MimiQ (Choi et al., 2025), the synthetic samples from MaskAQ (row 3 of Fig.4(a)) exhibit rich semantic diversity and more coherent structural information. Meanwhile, these synthetic samples are capable of focusing on larger object regions (*i.e.*, informative regions), which enable $Q$ to align with $P$, since low-precision $Q$ may not always be able to identify the correct regions due to large quantization error. Additionally, the synthetic samples from our MaskAQ retain sufficient semantics from $P$ (row 1 of Fig.4(b)), while maintain the attentional alignment between $P$ and $Q$ (row 2 of Fig.4(b)) across the task-relevant regions, which is consistent with *our proposal of coupling the selected informative regions of synthetic samples with varied Q in Sec.2.2*.

## 4. Conclusion and Limitations

This paper presents MaskAQ, a novel and effective approach for Data-Free Quantization of ViTs. Our core innovation lies in the insights of *informative regions*, which integrate the entropy-based region decoupling for semantic dispersion issue and the masked attention coupling for attentional disparity issue, to ensure high-quality sample synthesis. As a byproduct, a periodic refreshing enables MaskAQ's continuous adaptation to evolving $Q$. Across a wide range of backbones and downstream tasks (*i.e.*, classification, detection and segmentation), extensive experiments verify the superiority of MaskAQ over state-of-the-art approaches.

**Limitations**. The current MaskAQ still depends on iterative image synthesis, requiring additional generation overhead, while its applicability to more aggressive quantization settings remains to be explored in future work.

## Acknowledgements

This work was supported in part by Beijing Natural Science Foundation (No. L257005), and is also supported by National Natural Science Foundation of China (No. 92570204 and 62441235). This research is also partially supported by CCF-NetEase ThunderFire Innovation Research Funding (No. CCF-Netease 202513), and Institute of Advanced Medicine and Frontier Technology (2023IHM01080).

## Impact Statement

This paper presents work whose goal is to advance the field of Machine Learning. There are many potential societal consequences of our work, none of which we feel must be specifically highlighted here.

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

# A. Details of Related Work

As indicated in Sec.1 of the main body, we provide the comprehensive review of related work in this section, which mainly encompasses Vision Transformer architectures, data-driven quantization and data-free quantization approaches.

## A.1. Vision Transformer

The adaptation of the Transformer architecture from natural language processing to computer vision has reshaped visual recognition backbone design (Chen et al., 2021b; Wu et al., 2021). The original Transformer introduced multi-head self-attention as a general sequence modeling primitive, enabling models to capture long-range dependencies without recurrence or convolution. Building on that, Dosovitskiy et al. (Dosovitskiy, 2020) introduced the Vision Transformer (ViT), which processes images as sequences of patches. With large-scale pre-training, ViT competes with or outperforms convolutional networks on image classification, demonstrating that such pre-training can replace convolutional inductive biases. Subsequent work has improved ViT training efficiency and architectural suitability. DeiT (Touvron et al., 2021) introduced a data-efficient training method with token-based distillation, achieving competitive performance on ImageNet without massive external datasets and thus facilitating practical ViT adoption. Other lines of research modify the ViT architecture to better capture multi-scale and local visual structure. For example, Swin Transformer (Liu et al., 2021b) injects a hierarchical representation and shifted-window self-attention to achieve linear complexity with image size while retaining cross-window interactions. These advances establish it as a powerful general-purpose backbone for diverse computer vision tasks, *e.g.*, object detection (Carion et al., 2020), semantic segmentation (Chen et al., 2021a; Zheng et al., 2021) and video recognition (Arnab et al., 2021; Neimark et al., 2021). Therefore, the DFQ for ViTs are directly influenced by their architectural distinctions from Convolutional Neural Networks (CNNs), including patch-based tokenization, LayerNorm usage, attention-driven operations, and the distribution sensitivity of multi-head activations.

## A.2. Data-Driven Quantization for Vision Transformers

Data-driven quantization for Vision Transformers includes two complementary paradigms: quantization-aware training (QAT), where quantizers and weights are learned jointly on labeled data (Esser et al., 2020); and post-training quantization (PTQ), where a small unlabeled calibration set and reconstruction objectives are used to set quantization parameters without end-to-end retraining (Li et al., 2023b). Block-wise reconstruction methods such as BRECQ (Li et al., 2021) demonstrated that PTQ can be pushed to very low bit-widths (e.g., INT4/INT2) by reconstructing network blocks, inspiring follow-up PTQ work for transformers. Recent research (Li et al., 2023b; Jiang et al., 2025) in ViT post-training quantization has revealed that architectural components like LayerNorm and non-linear activations generate unique, non-uniform activation distributions—characterized by inter-channel variations, power-law patterns, and outliers—which necessitate specialized quantization strategies. To address these challenges, recent advances have introduced innovative solutions: the AdaLog framework (Wu et al., 2024) proposes adaptive logarithmic quantizers that dynamically optimize quantization parameters to better handle post-GELU and post-softmax activations, while APHQ-ViT (Wu et al., 2025) employs Hessian-aware reconstruction techniques with improved importance weighting to stabilize ultra-low precision (3-4 bit) quantization for ViTs.

However, both QAT and PTQ necessitate access to the original training data, which poses significant risks to data privacy and security in data-sensitive application scenarios.

## A.3. Data-Free Quantization for Vision Transformers

Data-free quantization (DFQ), which compresses models without original data, originated in the CNN domain. Early CNN-based methods (Xu et al., 2020; Choi et al., 2021; Zhong et al., 2022; Qian et al., 2023b;a; Dung et al., 2024; Kim et al., 2025) leveraged BatchNorm statistics to approximate data distributions. However, the patch-wise and self-attention mechanisms of ViTs violate core CNN assumptions, rendering them inadequate, thus motivating dedicated ViT-specific DFQ designs.

As a pioneer in ViT quantization, PSAQ-ViT (Li et al., 2022) and PSAQ-ViT V2 (Li et al., 2023a) synthesize calibration data by modeling patch relationships and attention responses, which introduces a patch-similarity objective to generate samples that mimic ViT interactions, thereby setting an early DFQ baseline for ViTs. Later work (Tong et al., 2025) advances this concept by proposing a curriculum-based synthesis pipeline that generates progressively challenging samples in a curriculum (easy-to-hard) manner. This strategy enhances quantization stability without the need for fine-tuning. A second line of work

prioritizes semantic and attention alignment. SARDFQ (Zhong et al., 2025) addresses the lack of semantic completeness in naive synthetic images by introducing attention-prior alignment and multi-semantic prompting, thereby ensuring calibration samples better represent key features and attention maps. Complementary to semantic prompting, MimiQ (Choi et al., 2025) observes that real images produce structured, inter-head attention similarity in ViTs. It optimizes synthetic samples to mimic these patterns and employs head-wise attention distillation, significantly boosting low-bit quantization performance.

Different from the prior work, our paper investigates the semantic dispersion and attentional disparity in the existing arts and aims to couple the selected informative regions of synthetic samples with varying Q, thus yielding high-quality synthetic samples.

## B. Proof Theorem 1 and Theorem 2

In this section, we further provide the detailed proofs of Theorem 1 and Theorem 2 in the main body.

### B.1. Preliminaries

**Notations**. We consider a classification setting with a discrete class label $y \in \mathcal{Y}$, where $\mathcal{Y}$ is a finite set. For each *real* sample, $a_r$ denotes its informative-region variable, supported on a finite alphabet $\mathcal{A}_r$. The full-precision model $P$ and the quantized model $Q$ produce discrete prediction outputs on the informative region of real samples, denoted by $z_p^r$ and $z_q^r$, respectively, supported on finite alphabets $\mathcal{Z}_p^r$ and $\mathcal{Z}_q^r$. For each *synthetic* sample $x_s$, $a_s$ denotes its informative-region variable supported on $\mathcal{A}_s$, and $Q$ produces the corresponding prediction output $z_q^s$ supported on $\mathcal{Z}_q^s$.

**Total variation distance.** For two distributions $P$ and $Q$ defined on the same finite alphabet $\Omega$, the total variation (TV) distance is

$$D_{\mathrm{TV}}(P, Q) \triangleq \frac{1}{2} \sum_{\omega \in \Omega} |P(\omega) - Q(\omega)|, \tag{27}$$

where a basic property is that TV distance does not increase under marginalization. Specifically, if $U$ is a marginal of $(U, V)$, then

$$D_{\mathrm{TV}}(P_U, Q_U) \leq D_{\mathrm{TV}}(P_{U,V}, Q_{U,V}). \tag{28}$$

To enable the TV distance between conditional distributions, we interpret

$$D_{\mathrm{TV}}\big(P(U \mid A), Q(U \mid A)\big) \leq \varepsilon \tag{29}$$

as the *averaged conditional TV*:

$$\sum_{a \in \mathcal{A}} P(A{=}a) \, D_{\mathrm{TV}}\big(P(U \mid A{=}a), Q(U \mid A{=}a)\big) \leq \varepsilon. \tag{30}$$

**Entropy continuity**. Define

$$\phi(\varepsilon; M) \triangleq h_2(\varepsilon) + \varepsilon \log(M - 1), \qquad h_2(\varepsilon) = -\varepsilon \log \varepsilon - (1 - \varepsilon) \log(1 - \varepsilon), \tag{31}$$

where log is the natural logarithm. We use the following standard continuity bound.

**Lemma 1** (Entropy continuity). *Let $U$ be a discrete random variable supported on an alphabet of size $M$. If $D_{\mathrm{TV}}(P_U, Q_U) \leq \varepsilon \leq \frac{1}{2}$, then*

$$|H_P(U) - H_Q(U)| \leq \phi(\varepsilon; M). \tag{32}$$

**Lemma 2** (Conditional entropy continuity). *Let $U$ and $A$ be discrete, with $U$ supported on an alphabet of size $|\mathcal{U}|$. If the averaged conditional TV satisfies*

$$\sum_a P(A{=}a) \, D_{\mathrm{TV}}\big(P(U \mid A{=}a), Q(U \mid A{=}a)\big) \leq \varepsilon \leq \frac{1}{2},$$

*then*

$$|H_P(U \mid A) - H_Q(U \mid A)| \leq \phi(\varepsilon; |\mathcal{U}|). \tag{33}$$

## B.2. Proof of Theorem 1

**Theorem 1** (Informative Region Alignment) *Let $a_r$ denote the informative-region variable of real samples, and let $z_p^r$ and $z_q^r$ be the corresponding prediction outputs of $P$ and $Q$, respectively. Assume that for some $\varepsilon_r \leq \frac{1}{2}$,*

$$D_{\mathrm{TV}}\Big(p_{\theta_q}(z_q^r, y \mid a_r),\ p_{\theta_p}(z_p^r, y \mid a_r)\Big) \leq \varepsilon_r, \tag{34}$$

$$D_{\mathrm{TV}}\Big(p_{\theta_q}(a_r, y \mid z_q^r),\ p_{\theta_p}(a_r, y \mid z_p^r)\Big) \leq \varepsilon_r, \tag{35}$$

*where $D_{\mathrm{TV}}(\cdot, \cdot)$ denotes the total variation (TV) distance. Then $Q$ approximately preserves $P$'s predictive information on informative regions in the sense that*

$$\big|I_{\theta_q}(z_q^r; y) - I_{\theta_p}(z_p^r; y)\big| \leq \Delta_r(\varepsilon_r), \tag{36}$$

*where $\Delta_r(\varepsilon_r)$ vanishes as $\varepsilon_r \to 0$.*

*Proof.* According to the chain rule, the mutual information can be decomposed as:

$$I_{\theta_q}(z_q^r; y) = I_{\theta_q}(z_q^r; y \mid a_r) + I(a_r; y) - I_{\theta_q}(a_r; y \mid z_q^r), \tag{37}$$

$$I_{\theta_p}(z_p^r; y) = I_{\theta_p}(z_p^r; y \mid a_r) + I(a_r; y) - I_{\theta_p}(a_r; y \mid z_p^r). \tag{38}$$

Subtracting Eq. (38) from Eq. (37) cancels the shared term $I(a_r; y)$, yielding

$$I_{\theta_q}(z_q^r; y) - I_{\theta_p}(z_p^r; y) = \underbrace{\Big(I_{\theta_q}(z_q^r; y \mid a_r) - I_{\theta_p}(z_p^r; y \mid a_r)\Big)}_{\triangleq \Delta_1} - \underbrace{\Big(I_{\theta_q}(a_r; y \mid z_q^r) - I_{\theta_p}(a_r; y \mid z_p^r)\Big)}_{\triangleq \Delta_2}. \tag{39}$$

By the triangle inequality, we have

$$\big|I_{\theta_q}(z_q^r; y) - I_{\theta_p}(z_p^r; y)\big| \leq |\Delta_1| + |\Delta_2|. \tag{40}$$

### Part 1: Bound $|\Delta_1|$ using Eq. (34)

We first expand conditional mutual information into conditional entropies below:

$$\begin{aligned}
|\Delta_1| &= \left|I_{\theta_q}(z_q^r; y \mid a_r) - I_{\theta_p}(z_p^r; y \mid a_r)\right| \\
&= \Big|[H_{\theta_q}(z_q^r \mid a_r) + H_{\theta_q}(y \mid a_r) + H_{\theta_q}(z_q^r, y \mid a_r)] \\
&\quad - [H_{\theta_p}(z_p^r \mid a_r) + H_{\theta_p}(y \mid a_r) + H_{\theta_p}(z_p^r, y \mid a_r)]\Big| \\
&\leq \left|H_{\theta_q}(z_q^r \mid a_r) - H_{\theta_p}(z_p^r \mid a_r)\right| + \left|H_{\theta_q}(y \mid a_r) - H_{\theta_p}(y \mid a_r)\right| \\
&\quad + \left|H_{\theta_q}(z_q^r, y \mid a_r) - H_{\theta_p}(z_p^r, y \mid a_r)\right|.
\end{aligned} \tag{41}$$

By using the assumption in Eq. (34)

$$D_{\mathrm{TV}}\Big(p_{\theta_q}(z_q^r, y \mid a_r),\ p_{\theta_p}(z_p^r, y \mid a_r)\Big) \leq \varepsilon_r$$

and the marginalization contraction of TV, we also have

$$D_{\mathrm{TV}}\Big(p_{\theta_q}(z_q^r \mid a_r),\ p_{\theta_p}(z_p^r \mid a_r)\Big) \leq \varepsilon_r, \qquad D_{\mathrm{TV}}\Big(p_{\theta_q}(y \mid a_r),\ p_{\theta_p}(y \mid a_r)\Big) \leq \varepsilon_r.$$

Hence, applying the conditional entropy continuity bound Eq. (33) to each term in Eq. (41) yields

$$|\Delta_1| \leq \phi(\varepsilon_r; |\mathcal{Z}_q^r|) + \phi(\varepsilon_r; |\mathcal{Y}|) + \phi(\varepsilon_r; |\mathcal{Z}_q^r||\mathcal{Y}|). \tag{42}$$

### Part 2: Bound $|\Delta_2|$ using Eq. (35)

Similarly, we can expand

$$
\begin{aligned}
|\Delta_2| &= \left| I_{\theta_q}(a_r; y \mid z_q^r) - I_{\theta_p}(a_r; y \mid z_p^r) \right| \\
&\leq \left| H_{\theta_q}(a_r \mid z_q^r) - H_{\theta_p}(a_r \mid z_p^r) \right| + \left| H_{\theta_q}(y \mid z_q^r) - H_{\theta_p}(y \mid z_p^r) \right| \\
&\quad + \left| H_{\theta_q}(a_r, y \mid z_q^r) - H_{\theta_p}(a_r, y \mid z_p^r) \right|.
\end{aligned}
\tag{43}
$$

By using the assumption in Eq. (35)

$$
D_{\mathrm{TV}}\Big( p_{\theta_q}(a_r, y \mid z_q^r),\ p_{\theta_p}(a_r, y \mid z_p^r) \Big) \leq \varepsilon_r
$$

and the marginalization contraction, we have

$$
D_{\mathrm{TV}}\Big( p_{\theta_q}(a_r \mid z_q^r),\ p_{\theta_p}(a_r \mid z_p^r) \Big) \leq \varepsilon_r, \qquad D_{\mathrm{TV}}\Big( p_{\theta_q}(y \mid z_q^r),\ p_{\theta_p}(y \mid z_p^r) \Big) \leq \varepsilon_r.
$$

Hence, applying Eq. (33) to each term in Eq. (43) yields

$$
|\Delta_2| \leq \phi(\varepsilon_r; |\mathcal{A}_r|) + \phi(\varepsilon_r; |\mathcal{Y}|) + \phi(\varepsilon_r; |\mathcal{A}_r||\mathcal{Y}|).
\tag{44}
$$

**Part 3**

Plugging Eq. (42) and Eq.(44) into Eq. (40) gives

$$
\begin{aligned}
\left| I_{\theta_q}(z_q^r; y) - I_{\theta_p}(z_p^r; y) \right| &\leq \phi(\varepsilon_r; |\mathcal{Z}_q^r|) + 2\phi(\varepsilon_r; |\mathcal{Y}|) + \phi(\varepsilon_r; |\mathcal{Z}_q^r||\mathcal{Y}|) \\
&\quad + \phi(\varepsilon_r; |\mathcal{A}_r|) + \phi(\varepsilon_r; |\mathcal{A}_r||\mathcal{Y}|).
\end{aligned}
\tag{45}
$$

Define the right-hand side as $\Delta_r(\varepsilon_r)$. Since $\phi(\varepsilon; M) \to 0$ as $\varepsilon \to 0$ for any fixed $M$, we conclude that $\Delta_r(\varepsilon_r) \to 0$ as $\varepsilon_r \to 0$.

$\square$

### B.3. Proof of Theorem 2

**Theorem 2** (Utility of Synthetic Sample) *Let $a_s$ denote the informative-region variable of synthetic samples $x_s$, and let $z_q^s$ be the corresponding prediction outputs of Q. If there exists $\varepsilon_s \leq \frac{1}{2}$ such that*

$$
D_{\mathrm{TV}}\Big( p_{\theta_q}(z_q^s, y \mid a_s),\ p_{\theta_p}(z_p^r, y \mid a_r) \Big) \leq \varepsilon_s,
\tag{46}
$$

$$
D_{\mathrm{TV}}\Big( p_{\theta_q}(a_s, y \mid z_q^s),\ p_{\theta_p}(a_r, y \mid z_p^r) \Big) \leq \varepsilon_s,
\tag{47}
$$

*and the region-label informativeness between synthetic and real informative regions is matched:*

$$
\left| I(a_s; y) - I(a_r; y) \right| \leq \xi,
\tag{48}
$$

*then synthetic samples enable Q to approximate P in predictive information:*

$$
\left| I_{\theta_q}(z_q^s; y) - I_{\theta_p}(z_p^r; y) \right| \leq \Delta_s(\varepsilon_s) + \eta + \Delta_a(\varepsilon_s),
\tag{49}
$$

*where $\Delta_s(\varepsilon_s)$ and $\Delta_a(\varepsilon_s)$ vanish as $\varepsilon_s \to 0$.*

*Proof.* According to the chain rule, we decompose the following

$$
I_{\theta_q}(z_q^s; y) = I_{\theta_q}(z_q^s; y \mid a_s) + I(a_s; y) - I_{\theta_q}(a_s; y \mid z_q^s),
\tag{50}
$$

$$
I_{\theta_p}(z_p^r; y) = I_{\theta_p}(z_p^r; y \mid a_r) + I(a_r; y) - I_{\theta_p}(a_r; y \mid z_p^r).
\tag{51}
$$

Subtract Eq. (51) from Eq. (50) and take absolute value, yielding

$$\left| I_{\theta_q}(z_q^s; y) - I_{\theta_p}(z_p^r; y) \right| \leq \underbrace{\left| I_{\theta_q}(z_q^s; y \mid a_s) - I_{\theta_p}(z_p^r; y \mid a_r) \right|}_{\triangleq T_1} + \underbrace{\left| I(a_s; y) - I(a_r; y) \right|}_{\triangleq T_2}$$
$$+ \underbrace{\left| I_{\theta_q}(a_s; y \mid z_q^s) - I_{\theta_p}(a_r; y \mid z_p^r) \right|}_{\triangleq T_3}. \tag{52}$$

By assumption Eq. (48), we have $T_2 \leq \xi$. It remains to bound $T_1$ and $T_3$.

**Part 1: Bound $T_1$ using Eq. (46)**

By expanding conditional mutual information into conditional entropies, we have

$$T_1 = \left| I_{\theta_q}(z_q^s; y \mid a_s) - I_{\theta_p}(z_p^r; y \mid a_r) \right|$$
$$\leq \left| H_{\theta_q}(z_q^s \mid a_s) - H_{\theta_p}(z_p^r \mid a_r) \right| + \left| H_{\theta_q}(y \mid a_s) - H_{\theta_p}(y \mid a_r) \right|$$
$$+ \left| H_{\theta_q}(z_q^s, y \mid a_s) - H_{\theta_p}(z_p^r, y \mid a_r) \right|. \tag{53}$$

Based on the assumption in Eq. (46)

$$D_{\text{TV}}\Big( p_{\theta_q}(z_q^s, y \mid a_s), \ p_{\theta_p}(z_p^r, y \mid a_r) \Big) \leq \varepsilon_s$$

and the marginalization contraction, they also imply

$$D_{\text{TV}}\Big( p_{\theta_q}(z_q^s \mid a_s), \ p_{\theta_p}(z_p^r \mid a_r) \Big) \leq \varepsilon_s, \qquad D_{\text{TV}}\Big( p_{\theta_q}(y \mid a_s), \ p_{\theta_p}(y \mid a_r) \Big) \leq \varepsilon_s.$$

Applying Eq. (33) term-wise in Eq. (53) yields

$$T_1 \leq \phi(\varepsilon_s; |\mathcal{Z}_q^s|) + \phi(\varepsilon_s; |\mathcal{Y}|) + \phi(\varepsilon_s; |\mathcal{Z}_q^s||\mathcal{Y}|) \triangleq \Delta_s(\varepsilon_s), \tag{54}$$

which leads to a vanishing error term $\Delta_s(\varepsilon_s) \to 0$ as $\varepsilon_s \to 0$.

**Part 2: Bound $T_3$ using Eq. (47)**

Similarly, we expand

$$T_3 = \left| I_{\theta_q}(a_s; y \mid z_q^s) - I_{\theta_p}(a_r; y \mid z_p^r) \right|$$
$$\leq \left| H_{\theta_q}(a_s \mid z_q^s) - H_{\theta_p}(a_r \mid z_p^r) \right| + \left| H_{\theta_q}(y \mid z_q^s) - H_{\theta_p}(y \mid z_p^r) \right|$$
$$+ \left| H_{\theta_q}(a_s, y \mid z_q^s) - H_{\theta_p}(a_r, y \mid z_p^r) \right|. \tag{55}$$

Assumption Eq. (47) states

$$D_{\text{TV}}\Big( p_{\theta_q}(a_s, y \mid z_q^s), \ p_{\theta_p}(a_r, y \mid z_p^r) \Big) \leq \varepsilon_s.$$

By marginalization contraction, it implies

$$D_{\text{TV}}\Big( p_{\theta_q}(a_s \mid z_q^s), \ p_{\theta_p}(a_r \mid z_p^r) \Big) \leq \varepsilon_s, \qquad D_{\text{TV}}\Big( p_{\theta_q}(y \mid z_q^s), \ p_{\theta_p}(y \mid z_p^r) \Big) \leq \varepsilon_s.$$

Applying Eq. (33) term-wise in Eq. (55) yields

$$T_3 \leq \phi(\varepsilon_s; |\mathcal{A}_s|) + \phi(\varepsilon_s; |\mathcal{Y}|) + \phi(\varepsilon_s; |\mathcal{A}_s||\mathcal{Y}|) \triangleq \Delta_a(\varepsilon_s), \tag{56}$$

where $\Delta_a(\varepsilon_s) \to 0$ as $\varepsilon_s \to 0$.

**Part 3**

Plugging $T_1 \leq \Delta_s(\varepsilon_s)$ from Eq. (54), $T_2 \leq \xi$ from Eq. (48), and $T_3 \leq \Delta_a(\varepsilon_s)$ from Eq. (56) into Eq. (52), we conclude

$$\left| I_{\theta_q}(z_q^s; y) - I_{\theta_p}(z_p^r; y) \right| \leq \Delta_s(\varepsilon_s) + \xi + \Delta_a(\varepsilon_s).$$

Since $\phi(\varepsilon; M) \to 0$ as $\varepsilon \to 0$ for fixed $M$, both $\Delta_s(\varepsilon_s)$ and $\Delta_a(\varepsilon_s)$ vanish as $\varepsilon_s \to 0$.

$\square$

*Table 3.* Comparison with state-of-the-art approaches on the COCO dataset and ADE20K dataset. *m*w*n*a indicates the weights and activations are quantized to *m*-bit and *n*-bit. The best results are reported with **boldface**.

| Bit width | Methods | Venue | COCO Dataset | | ADE20K Dataset |
| --- | --- | --- | --- | --- | --- |
| | | | $AP_{box}$ | $AP_{mask}$ | mIoU |
| *4w4a* | PSAQ-ViT (Li et al., 2022) | ECCV' 22 | 0.06 | 0.06 | 1.65 |
| | PSAQ-ViT V2 (Li et al., 2023a) | T-NNLS' 23 | 4.52 | 5.03 | 3.83 |
| | MimiQ (Choi et al., 2025) | AAAI' 25 | 26.41 | 26.63 | 29.92 |
| | **MaskAQ (Ours)** | - | **27.11** | **27.19** | **31.04** |
| *5w5a* | PSAQ-ViT (Li et al., 2022) | ECCV' 22 | 0.41 | 0.46 | 20.26 |
| | PSAQ-ViT V2 (Li et al., 2023a) | T-NNLS' 23 | 32.69 | 31.21 | 26.35 |
| | MimiQ (Choi et al., 2025) | AAAI' 25 | 41.63 | 38.53 | 38.88 |
| | **MaskAQ (Ours)** | - | **42.55** | **38.96** | **40.12** |

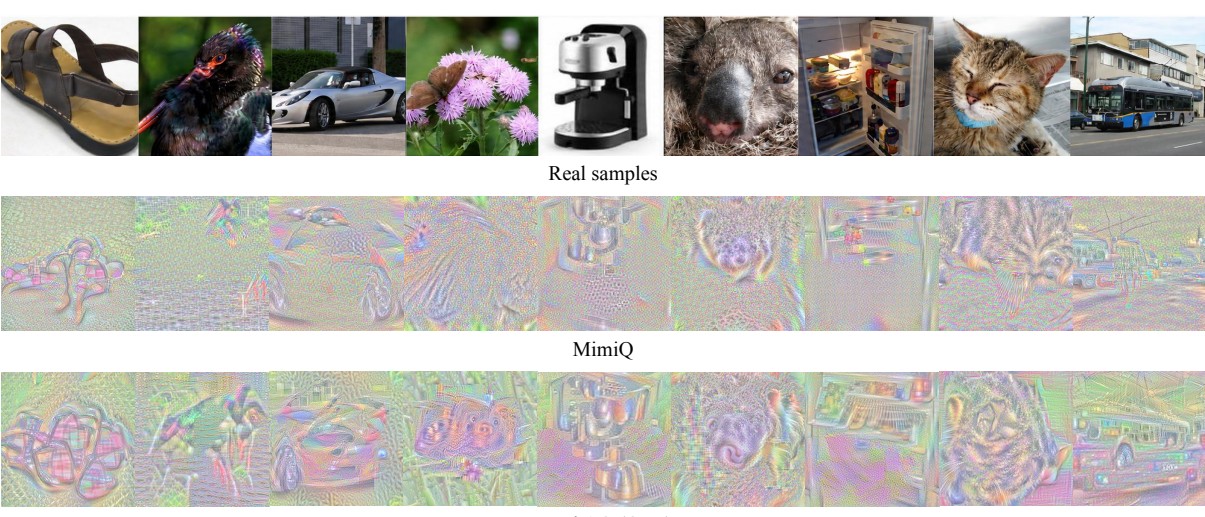

Real samples

MimiQ

MaskAQ (Ours)

*Figure 5.* More examples of synthetic samples from MimiQ (Choi et al., 2025) and our MaskAQ, apart from the corresponding real samples.

## C. Additional Discussions and Results

As indicated in Sec.3.1 of the main body, we further offer additional discussions and more experimental results.

### C.1. Additional Results on Object Detection and Semantic Segmentation Tasks

Apart from the image classification task in the main body, we further report the experimental results on object detection and semantic segmentation tasks. Specifically, we conduct object detection experiments with Swin-T backbone on the COCO dataset (Lin et al., 2014), which is a widely used benchmark with rich instance-level annotations across diverse scenes. Besides, we further evaluate semantic segmentation with Swin-T backbone on the ADE20K dataset (Zhou et al., 2017), which provides high-quality pixel-wise annotations over a broad set of semantic categories and scene types. For COCO, we report COCO-style bounding-box ($AP_{box}$) and instance-mask ($AP_{mask}$), where AP is averaged over IoU thresholds 0.50:0.05:0.95 and over all classes; while for ADE20K, Mean Intersection over Union (mIoU) serves as the evaluation metrics. Notably, for PSAQ-ViT (Li et al., 2022) and PSAQ-ViT V2 (Li et al., 2023a), we adopt the results reported in MimiQ (Choi et al., 2025). Table 3 suggests that our MaskAQ achieves great improvements over other approaches. For example, MaskAQ exhibits at most 0.92 and 1.24 gains on the COCO and ADE20K datasets, compared to state-of-the-art approaches, *e.g.*, MimiQ. The above fact further confirms the *broad compatibility* of MaskAQ on other tasks.

## C.2. Visualization of More Synthetic Samples

In Sec.3.4 of the main body, we visualize the synthetic samples from the existing arts and our MaskAQ. In this section, we offer more visual results of synthetic samples from MimiQ (Choi et al., 2025) and our MaskAQ, apart from the corresponding real samples, as illustrated in Fig.5. The results show that, the synthetic samples (row 3 of Fig.5) from MaskAQ exhibit rich semantic diversity and more coherent structural information. Meanwhile, these synthetic samples are capable of focusing on larger object regions (*i.e.*, informative regions), which enable $Q$ to align with $P$, since low-precision $Q$ may not always be able to identify the correct regions due to large quantization error. Additionally, we visualize the attention maps from the intermediate layer of $P$ and $Q$, as illustrated in Fig.6. It is observed that the synthetic samples from our MaskAQ retain sufficient semantics from $P$, while maintain the attentional alignment between $P$ and $Q$ (row 2 and 3 of Fig.6) across the task-relevant regions. The above fact confirms *the effectiveness of coupling the selected informative regions to yield high-quality synthetic samples*.

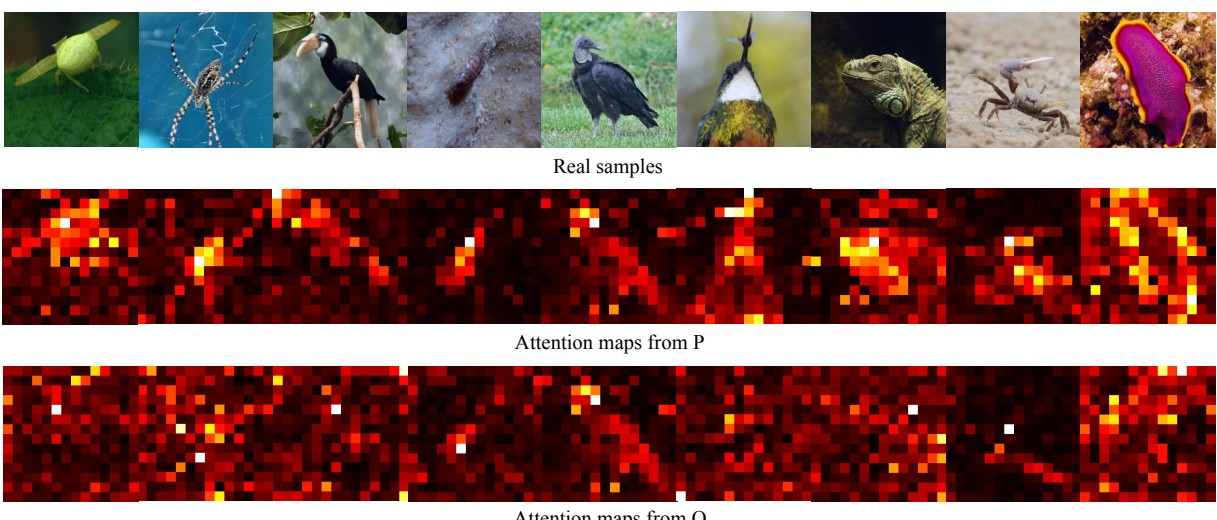

Real samples

Attention maps from P

Attention maps from Q

*Figure 6.* Additional visualization of attention maps (the brighter, the larger) from the intermediate layer of $P$ and $Q$ based on MaskAQ.

