# OpenReview forum: "Selective Coupling of Decoupled Informative Regions: Masked Attention Alignment for Data-Free Quantization of Vision Transformers"
_ICML.cc/2026/Conference — ICML 2026 regular_

### Official Review · Reviewer_JEde · 2026-02-23

**Soundness:** 2
**Presentation:** 3
**Significance:** 2
**Originality:** 2
**Overall Recommendation:** 3
**Confidence:** 5

**Summary:**

The paper presents MaskAQ, a data-free quantization method for Vision Transformers that focuses on enhancing synthetic calibration samples by targeting "informative regions". The proposed framework isolates these informative patches using an entropy regularizer on attention similarities , and utilizes a masked alignment loss to limit the agreement between P and Q strictly to these adaptively selected areas. A periodic refreshing strategy is also included to update the synthetic samples during training. The authors evaluate the method on ImageNet across several transformer backbones (ViT, DeiT, Swin) and report consistent accuracy gains, highlighting a maximum Top-1 improvement of 3.1% in the 3-bit setting.

**Compliance With Llm Reviewing Policy:**

Affirmed.

**Final Justification:**

I have reviewed the rebuttal and the authors' responses to my concerns. The additional experimental results provided have partially addressed my initial issues regarding the paper's soundness. Accordingly, I have increased my score from 2 (Reject) to 3 (Weak Reject).

**Key Questions For Authors:**

1. What are the training compute costs (GPU hours) for synthesis+calibration per model, and how do they compare to MimiQ and PSAQ-ViT? How do performance and cost vary with the number of synthetic images and refresh frequency?
2. For the entropy regularizer, how is $σ_l^2$ computed in practice? Did you empirically verify that the similarity distributions are indeed approximately Gaussian, as assumed in Eq. (8)?
3. The CLAMP-ViT results appear under different experimental settings but are directly compared in Table 2. Can you provide a unified comparison (same quantization, optimizer, epochs, and augmentations), or else clarify and relocate cross-setting results to avoid confusion?
4. Regarding Table 1, why is the comparison with GenQ (Li et al., 2024) restricted solely to the 4w4a setting and omitted from the 3w3a, 5w5a, and 4w8a configurations? Notably, GenQ outperforms the proposed MaskAQ on DeiT-B under 4w4a (76.10% vs. 74.41%), serving as a strong state-of-the-art baseline. The exclusion of this specific baseline in the other bit-width evaluations is highly conspicuous. Please provide the comparative results for GenQ across the remaining bit-widths, or provide a rigorous technical justification for its omission.

**Limitations:**

The authors have not adequately discussed the limitations or potential negative societal impacts of their work. To improve the manuscript, the following points must be explicitly addressed:
1. The authors should quantify the extra training costs (e.g., GPU hours) introduced by the adaptive mask generation and periodic sample refreshing compared to baselines.
2. The paper must discuss its sensitivity to the newly introduced hyperparameters ($p_{drop}$, $k_{min}$, $\lambda_{fb}$, $\lambda_{align}$) and acknowledge the uncertainty of scaling these results from tiny models (e.g., DeiT-T) to larger architectures.
3. Rather than a generic dismissal, the authors should briefly acknowledge standard risks associated with model quantization, such as the potential amplification of algorithmic biases in the compressed model.

**Strengths And Weaknesses:**

**Strengths**
1. Introduces a principled “informative region” concept in the DFQ-for-ViT setting and leverages it both in synthesis (entropy on attention similarity) and in calibration (masked alignment and masked loss weighting).
2. The masked attention alignment objective is a targeted alternative to holistic attention alignment, focusing the supervision where Q is most likely to benefit and avoiding over-regularizing background.
3. The breadth across models and bit-widths is a plus, and gains over MimiQ are consistent. The 3-bit DeiT-T gain (+3.1%) is noteworthy given the difficulty of low-bit DFQ for ViTs.

**Weaknesses**
1. The entropy-based informative-region decoupling is intuitively plausible but mathematically under-specified. Moving from a discrete histogram of pairwise attention-vector similarities to a Gaussian differential entropy surrogate is a significant modeling shortcut. A more rigorous justification (e.g., empirical validation or an ablation comparing histogram entropy versus the Gaussian proxy) is necessary.
2. The fairness of the CLAMP-ViT comparison is questionable due to different settings. Presenting both methods under a unified protocol, or explicitly segregating these cross-setting results from the main table, is required for a rigorous evaluation.
3. Critical missing implementation details hinder reproducibility. While the stochastic mask dropping is claimed as a core contribution, the exact values for hyper-parameters such as $p_{drop}$ and $k_{min}$ in Eq. (11) are nowhere to be found in Section 3.1.
4. The ablation studies (Table 3)  are conducted exclusively on extremely small models (ViT-T and DeiT-T). It remains unclear whether the individual components of MaskAQ scale effectively and maintain their relative importance on larger capacity models like DeiT-S or DeiT-B.
5. The overall conceptual contribution feels incremental rather than groundbreaking. Prior arts (e.g., PSAQ-ViT, MimiQ) have already established the paradigm of leveraging patch similarity and attention maps for ViT sample synthesis. While introducing entropy for background decoupling and masking the alignment objective are reasonable engineering improvements, they are marginal extensions of existing ideas rather than a fundamental shift in tackling DFQ for ViTs.

---

> ### Author Rebuttal · Authors · 2026-03-31
>
> We sincerely thank Reviewer JEde for the constructive comments.  Below are how we address your major concerns due to space limitation.
>
> ---
> ### **Q1**:  About the entropy-based term.
> **A1**: We appreciate this comment. We kindly clarify that Eq. (8) is not a strict equivalence, but serves as a smooth and tractable surrogate for entropy maximization. In practice, $\delta_l^2$ is computed as the empirical variance of all off-diagonal pairwise cosine similarities $S_{ij}, i,j=1,2,...,N, i \neq j$ at layer $l$, namely:
>
> $\delta_l^2=\frac{1}{M}\sum_{i\ne j}\left(S_{ij}-\mu_l\right)^2$,
>
> $\mu_l=\frac{1}{M}\sum_{i\ne j} S_{ij}, M=N(N−1)$.
>
> We adopt the Gaussian differential entropy for optimization stability, since the histogram estimator introduces binning sensitivity and less stable gradients during synthesis, in line with prior work, e.g., PSAQ-ViT.
>
>
>
> ---
> ### **Q2**: Lack the training cost analysis.
> **A2**: Thanks for pointing this out. As you suggested, we add more experiments with DeiT-S under 3-bit cases below:
> | Method | Synthesis Time  | Calibration Time  | Acc. (%) |
> |--------|------|----------|----------|
> | PSAQ-ViT  | 0.2h |  0.0005h   |    0.14  |
> | MimiQ  | 12.2h | 2.1h   |  27.39  |
> | MaskAQ  |13.8h  | 2.1h   |  30.41   |
>
> | Sample Number |2k | 5k | 8k | 10k|
> |--------|------|------|------|------|
> | Acc. (%)  |24.32 |28.55 | 29.80 |**30.41**  |
>
> The results reveal that MaskAQ incurs additional cost mainly in the synthesis stage, while the calibration stage remains in the same order as prior DFQ work, which is consistent with Sec.2.2 and 2.3.
>
> ---
> ### **Q3**: A unified comparison of CLAMP-ViT results.
> **A3**: Thanks for the comments. Actually, the CLAMP-ViT results in Table 2 were reproduced under **the same settings** as MimiQ and our MaskAQ below:
> | | Backbone | CLAMP-ViT | MimiQ | MaskAQ |
> |---|---|---|---|---|
> | 4w4a | DeiT-T | 0.15 | 52.03 | **53.01** |
> | | DeiT-S | 0.13 | 62.72 | **63.81** |
> | 5w5a | DeiT-T | 5.42 | 63.40 | **64.27** |
> | | DeiT-S | 3.19 | 72.59 | **73.18** |
> | 4w8a | DeiT-T | 65.71 | 69.86 | **69.95** |
> | | DeiT-S | 76.44 | 78.48 | **79.44** |
>
> As you suggested, we will merge Table 2 into Table 1 to avoid confusion.
>
> ---
> ### **Q4**: Missing implementation details.
> **A4**: Thanks for pointing this out. In our experiments, $k_{min}=1$ serves as a safeguard to avoid dropping all patches, exhibiting no meaningful sensitivity in training, while $p_{drop}=0.3$ introduces a moderate level of stochasticity.
> | $p_{drop}$ | 0.1 | 0.2 | 0.3 | 0.4 | 0.5 |
> |--------|------|------|------|------|------|
> | Acc. (%)  | 43.04 | 43.13 | **43.28** | 43.18  | 42.93 |
>
> The results validate that $p_{drop}$  is robust to its values, since it mainly introduces mild stochasticity, rather than a sensitive performance-critical hyperparameter.
>
> ---
> ### **Q5**: The ablation studies (Table 3) are conducted exclusively on extremely small models.
> **A5**:  As you suggested, we conducted the same ablation study on DeiT-S / DeiT-B under 3-bit cases below:
> | Cases | DeiT-S | DeiT-B |
> |--------|------|------|
> | w/o $L_{fb}$  |28.82  |42.62  |
> | w/o $L_{align}$  |28.60  |42.23  |
> | w/o Refreshing   |29.54  |42.87  |
> | w/o $L_{Q}$  | 29.96 | 43.04  |
> | MaskAQ  | **30.41**  | **43.28**  |
>
> The results show that the individual components not only scale to larger-capacity backbones, but also maintain broadly similar relative importance.
>
> ---
> ### **Q6**: The overall conceptual contribution.
> **A6**: We kindly clarify the followings:
> - **Our core contribution** is to tackle DFQ for ViT from **a novel information bottleneck perspective** (see **theoretical analysis in Sec.2.4**): *the mutual information dominates the primary principle of sample synthesis to crucially associate informative regions of synthetic samples with Q* (**Line 250-274**), rather than merely improving global realism or generic attention consistency in prior DFQ work (e.g., PSAQ-ViT, MimiQ) .
> - To resolve **how the synthetic samples preserve crucial information for Q’s calibration** (**Line 99-102**), the unified calibration-oriented synthesis is proposed to identify and align the sparse informative regions that dominate calibration-relevant semantics for Q (**Sec. 2.2** and **2.3**), which is a *fundamental shift in synthesis principle*.
>
> ---
> ### **Q7**: About the comparison with GenQ.
> **A7**: We mildly clarify that, GenQ only performs under 4w4a, while **missed on other settings with no code available**.  To avoid potential unfair re-implementation, we restricted GenQ to the configuration explicitly reported by the original paper. To be comprehensive, we compare with all other competitors across all other settings  (e.g., 5w5a and 4w8a for PSAQ-ViT and MimiQ, 3w3a for MimiQ) in **Table 1 and 2**, where MaskAQ outperforms these competitors ranging from low-to-high bit settings (**Line 370-377**).
>
> ---
> ### We are sincerely grateful to your questions, while appreciate your efforts by taking our above responses into final consideration.

---

> > ### Author Rebuttal · Reviewer_JEde · 2026-04-01
> >
> > I thank the authors for their response and the additional experiments.
> >
> > **Partially addressed concerns:**
> >
> > The CLAMP-ViT comparison has been clarified (Q3). Merging Table 2 into Table 1 is the right call. The ablation on DeiT-S and DeiT-B (Q5) shows that individual components scale to larger models, which was a gap in the original submission. The training cost breakdown (Q2) and δ sensitivity analysis (Q4) are also appreciated.
> >
> > **Unresolved concerns:**
> >
> > Regarding the entropy regularizer (Q1): I specifically asked for an empirical comparison between histogram entropy and the Gaussian surrogate. The response instead explains why the Gaussian proxy was chosen (optimization stability, binning sensitivity) and appeals to prior work. This does not replace an actual ablation showing that the surrogate is a reasonable approximation in practice. Whether the similarity distributions are approximately Gaussian remains unverified.
> >
> > Regarding GenQ (Q7), the justification for restricting comparison to 4w4a is reasonable given the absence of released code. That said, the 3w3a evaluation still only compares against MimiQ, while other baselines (GDFQ, Qimera, AdaDFQ, PSAQ-ViT, PSAQ-ViT V2) are all present in 4w4a and 5w5a. The 3w3a setting is the most challenging and where the proposed method's value should be most apparent. I would encourage the authors to add these comparisons.
> >
> > Regarding novelty (Q6): The authors reframe their contribution as an "information bottleneck perspective" and argue this constitutes a fundamental shift from prior DFQ work. I am not convinced. The actual method consists of an entropy regularizer on attention similarities, masked alignment loss, and periodic sample refreshing. These are reasonable engineering improvements on top of the PSAQ-ViT/MimiQ paradigm, but they do not represent a qualitative departure from the existing approach of leveraging patch similarity and attention maps for ViT sample synthesis. The theoretical analysis in Sec. 2.4 formalizes the motivation but does not change the nature of the technical contribution itself.

---

> > > ### Author Response · Authors · 2026-04-02
> > >
> > > We appreciate your timely follow-up and the recognition of our additional experiments. Below are our further responses to the remaining concerns.
> > >
> > > ---
> > >
> > > ## **Response to Regarding the entropy regularizer (Q1)**
> > > Thanks for your feedback. We apologize for omitting some information in our previous round, owing to the tight time. Following your insightful suggestions, we conduct the ablation comparison between histogram entropy and Gaussian surrogate with DeiT-S and DeiT-B under 3-bit cases below:
> > >
> > > | Cases | DeiT-S  | DeiT-B  |
> > > |---|---|---|
> > > | w/o $L_{fb}$ | 28.64$\pm$0.23 | 42.37$\pm$0.28 |
> > > | $L_{fb}$ w/ Histogram entropy | 30.09$\pm$0.19 |42.96$\pm$0.26 |
> > > |$L_{fb}$ w/ Gaussian surrogate | 30.37$\pm$0.06 | 43.21$\pm$0.10 |
> > >
> > > The results suggest that Gaussian surrogate preserves similar or slightly better final performance, while improving **optimization stability** and avoiding binning sensitivity, compared to histogram entropy.
> > >
> > > ---
> > >
> > > ## **Response to Regarding 3w3a setting (Q7)**
> > > Thanks for raising this question. We kindly clarify the followings:
> > > - In our re-implementation, other prior baselines (e.g., GDFQ, Qimera, AdaDFQ, PSAQ-ViT, PSAQ-ViT V2) at 3w3a became **highly unstable** and typically **collapsed** to near-random accuracy (below 5%, or even lower), making them **uninformative for meaningful comparison**.
> > > - Notably, such trend is already visible at **4w4a** (refer to **Table 1**), where several prior baselines are already close to **instability**, suggesting that 3w3a is a failure regime for most of prior baselines, as shown below:
> > >
> > > | Method | ViT-T  | ViT-B  | DeiT-T  | DeiT-S  | DeiT-B  | Swin-T |
> > > |---|---:|---:|---:|---:|---:|---:|
> > > | GDFQ | `2.95` | `11.73` | 25.96 | 22.12 | 30.04 | 42.08 |
> > > | Qimera | `0.57` | `5.61` | `15.18` | `11.37` | 32.49 | 47.98 |
> > > | AdaDFQ | `2.00` | `6.21` | 19.57 | `14.44` | 19.22 | 38.88 |
> > > | PSAQ-ViT | `0.67` | `0.94` | 19.61 | `5.90` | `8.74` | 22.71 |
> > > | PSAQ-ViT V2 | `1.54` | `2.83` | 22.82 | 32.57 | 45.81 | 50.42 |
> > > | GenQ  | - | 67.50 | - | - | 76.10 | - |
> > > | MimiQ  | 42.99 | 62.91 | 52.03 | 62.72 | 74.10 | 69.33 |
> > > | **MaskAQ (Ours)** | 44.90 | 70.53 | 53.10 | 63.81 | 74.41 | 70.40 |
> > >
> > > - In this sense, the fact that MaskAQ remains superior under 3w3a further confirms **our innovation of tackling the crucial challenge of DFQ for ultra-low-bit ViT**.
> > >
> > > ---
> > >
> > > ## **Response to Regarding novelty (Q6)**
> > > Thanks for your time and critical engagement with our work. We would like to further clarify that the core novelty of MaskAQ is to **shift the goal of DFQ sample synthesis from approximating real data (the prior work) to preserve the crucial information needed for calibrating Q**, so as to resolve the semantic dispersion and attentional disparity issues (see **Figure 1**). Specifically, we revisit DFQ for ViT from a **novel information bottleneck (IB) perspective** in Sec 2.4 below:
> > > $\max I(z_q; y), \text{s.t.} \ I(x; z_q) \leq C$ in Eq.(16),
> > > where the synthetic samples must preserve the subset of information that remains most useful for Q under *limited bit capacity*, motivating the followings:
> > >
> > > - **Decoupling informative regions.** The prior arts utilize attention maps as a heuristic proxy for visual realism, essentially trying to reconstruct real-data distribution. **Our entropy regularizer** is not a smoothing tool but a mathematical filter designed to purge "low-utility" background noise, hence the limited bit-capacity of the quantized model (Q) focuses exclusively on **high-entropy** informative regions, alleviating *semantic dispersion* issue (**Line 252-274**).
> > >
> > > - **Coupling informative regions with Q's capacity**. The prior work largely operates on global patch relations or overall attention structures for only full-precision model P, which **deviates from Q's calibration**, suffering from *attentional disparity* between P and Q. Our IB analysis requires the synthetic samples to derive calibration-relevant semantics for Q below:
> > > $D_{\mathrm{TV}}\Big(p_{\theta_q}(z_q^s,y\mid a_s),\ p_{\theta_p}(z_p^r,y\mid a_r)\Big)\le \varepsilon_s$ in Eq.(20),
> > > $D_{\mathrm{TV}}\Big(p_{\theta_q}(a_s,y\mid z_q^s),\ p_{\theta_p}(a_r,y\mid z_p^r)\Big)\le \varepsilon_s$ in Eq.(21).
> > > Anchored on informative regions, **our masked alignment loss** strictly enforces the bounds in Eq. (20) and (21), ensuring that the synthetic samples are aligned with the current capacity of Q (**Line 275-283**).
> > >
> > > **Dynamic synthesis for the evolving Q.** Instead of the **static** synthesis for only P, our periodic refreshing is a theoretical mandate to maintain the mutual information lower bound between the data and the **evolving** Q, ensuring the synthesis remains "**calibration-reliable**" throughout the training.
> > >
> > > Therefore, these components are not independent engineering add-ons, but **natural consequences** of the unified theoretical blueprint.
> > >
> > > ---
> > >
> > > ### We sincerely thank the reviewer again for your time and the efforts by taking our further responses into final consideration.

---

### Official Review · Reviewer_dhFR · 2026-03-11

**Soundness:** 3
**Presentation:** 3
**Significance:** 3
**Originality:** 3
**Overall Recommendation:** 4
**Confidence:** 3

**Summary:**

The paper proposes MaskAQ, a novel Data-Free Quantization (DFQ) method tailored for Vision Transformers (ViTs). The authors identify that existing ViT DFQ methods suffer from two primary issues: semantic dispersion (semantics distributed across the entire image) and attentional disparity (the quantized model struggles to align with the full-precision model). To resolve these issues, MaskAQ decouples "informative regions" from noisy backgrounds by maximizing the differential entropy of patch similarity. Furthermore, it employs an adaptive mask and a masked attention alignment objective to tightly couple the synthetic samples with the quantized model Q. The authors also introduce a periodic sample refreshing strategy to adapt to the evolving state of Q during calibration. Extensive experiments demonstrate state-of-the-art performance on ImageNet classification, as well as COCO detection and ADE20K segmentation, particularly in low bit-width settings.

**Compliance With Llm Reviewing Policy:**

Affirmed.

**Final Justification:**

Most of my concerns are resolved. Experiment results show the effectiveness of indicating and utilizing informative regions in MaskAQ.

**Key Questions For Authors:**

1. Could you provide a quantitative comparison of the total wall-clock time and memory footprint required for the data synthesis and calibration phases of MaskAQ versus primary baselines (e.g., MimiQ)?
2. Could you provide a quantitative comparison of the total wall-clock time and memory footprint required for DFQ methods versus quantization? Because if such post-quantization methods consume more time and memory than quantization, why not use the original model P directly, since you also assume users have access to P?
3. Does the proposed Information Bottleneck theory hold robustly under lower bit-width than 3w3a? How does this extreme restriction practically affect the pattern of all the loss terms (L_prior, L_fb, L_align, L_S and L_Q) compared with higher precision?

**Limitations:**

Please see Weaknesses.

**Strengths And Weaknesses:**

Strengths
1. Identifying "semantic dispersion" and "attentional disparity" as key bottlenecks in generating synthetic calibration data for ViTs is insightful and supported by visual evidence.
2. The paper elevates its empirical design with theoretical analysis from an Information Bottleneck (IB) perspective. Section 2.4 proves that aligning informative regions within synthetic samples is mathematically sufficient to preserve the predictive information of the full-precision model.
3. MaskAQ achieves substantial performance improvements over a recent SOTA method: up to a 3.1% Top-1 accuracy gain over MimiQ on ImageNet for DeiT-T under 3w3a quantization. The consistent gains across multiple backbones (ViT, DeiT, Swin-T) and downstream tasks (detection, segmentation) prove the method's robust generalization capabilities.

Weaknesses
1. Missing computational overhead analysis: In Algorithm1, the proposed framework incorporates a periodic sample refreshing strategy throughout training and multiple loss terms computed throughout training. However, the paper lacks a comparative analysis of the wall-clock time and memory overhead required for this iterative synthesis and calibration phase compared to baselines like MimiQ or PSAQ-ViT, as well as the cost of quantization itself.
2. There are many hyperparameters within each loss term and additional ones for training. While the authors successfully ablate the balancing parameters λ_fb and λ_align, the adaptive mask generation introduces several other critical hyperparameters, including the dropout probability p_drop, the predefined minimum k_min, and the dynamic annealing schedule for k (reducing from 50% to 10%). The sensitivity of the model to these other specific masking variables is not fully explored.

---

> ### Author Rebuttal · Authors · 2026-03-31
>
> We sincerely thank Reviewer dhFR for the constructive comments.  Below are how we address your concerns and provide further clarification.
>
> ---
>
> ### **Q1**:  Missing computational overhead analysis.
>
> **A1**: Thanks for your comments. As you suggested, we added the computational overhead analysis with DeiT-S under 3-bit cases below:
>
> | Method | Synthesis Time  | Calibration/quantization Time  | GPU Memory (GB) | Acc. (%) |
> |--------|------|----------|----------|----------|
> | PSAQ-ViT  | 0.2h |  0.0005h   |4.6 |    0.14  |
> | MimiQ  | 12.2h | 2.1h   | 9.7|  27.39  |
> | MaskAQ  (Ours)  |13.8h  | 2.1h   | 10.5 |  30.41   |
>
> MaskAQ incurs additional cost mainly in the synthesis stage due to the extra entropy/alignment objectives and periodic refreshing, while the calibration stage remains in the same order as prior DFQ work. Importantly, these overheads are incurred *only during the offline DFQ process* and **do not affect** the inference-time cost of the final quantized model. Given the consistent accuracy gains, we believe this trade-off is reasonable. The above findings are complementary to our intuition, as indicated in Sec.1.
>
> ---
>
> ### **Q2**:  About the sensitivity of other specific masking hyperparameters.
>
> **A2**: Thanks for your comments. We kindly clarify that these masking hyperparameters **are not intended as fragile tuning knobs**, but as *coarse control parameters* for the exploration-to-refinement dynamics of informative-region selection. Following your suggestions, we added more ablation results with DeiT-B backbone under 3-bit cases below:
>
> | $p_{drop}$ | 0.1 | 0.2 | 0.3 | 0.4 | 0.5 |
> |--------|------|------|------|------|------|
> | Acc. (%)  | 43.04 | 43.13 | **43.28** | 43.18  | 42.93 |
>
> | Mask ratio (k) | Acc. (%) |
> |--------|------|
> | 50%  | 41.95 |
> | 10%  | 42.53 |
> | 50%-10%  | **43.28**  |
>
> The above results disclosed that the adaptive mask is *robust* to the choice of $p_{drop}$ and the dynamic schedule from 50% to 10% consistently *outperforms* fixed masking ratio, which further validate the intuition of our masked attention alignment, as indicated in **Line 191-219**. In particular, k_min=1 serves as a safeguard in training to avoid dropping all patches, hence exhibits *no meaningful sensitivity* in practice.
>
> ---
>
> ### **Q3**:  Could you provide a quantitative comparison for DFQ methods versus quantization? Why not use the original model P directly, since you also assume users have access to P?
>
> **A3**: Thanks for this comment. We clarify that DFQ methods incur wall-clock time and memory overhead only during the offline synthesis/calibration stage (**one-time offline cost**) below:
> | Method | Synthesis Time  | Calibration/quantization Time  | GPU Memory (GB) | Acc. (%) |
> |--------|------|----------|----------|----------|
> | PSAQ-ViT  | 0.2h |  0.0005h   |4.6 |    0.14  |
> | MimiQ  | 12.2h | 2.1h   | 9.7|  27.39  |
> | MaskAQ  (Ours)  |13.8h  | 2.1h   | 10.5 |  30.41   |
>
> The results suggest that DFQ methods require one-time affordable cost, yielding a substantially cheaper model for long-term deployment. In contrast, users may have access to P, but still cannot afford to deploy P directly due to memory or latency constraints (**repeated online deployment cost**).
>
> ---
>
> ### **Q4**:  Does the proposed Information Bottleneck theory hold robustly under lower bit-width than 3w3a? How does this extreme restriction practically affect the pattern of all the loss terms compared with higher precision?
>
> **A4**: Thanks for this insightful question. We kindly clarify the followings:
> - Below 3w3a, the information bottleneck view remains valid as a **mechanism-level explanation**: as the bit-width is *lower*, the retained information is more severely *compressed*, and the distribution mismatch between synthetic samples and inputs expected by Q becomes *larger*, making the bottleneck effect even more *pronounced* (**Line 252-274**).
> - The main difference under lower bit-widths lies in the **optimization difficulty** and **relative roles** of the loss terms, rather than the invalidation of the underlying intuition. Specifically, L_prior and L_fb become more critical for stabilizing the synthetic sample distribution and separating informative regions; while L_align and L_Q typically become larger and harder to optimize, owing to the amplified discrepancy between P and lower-bit Q (**Line 275-283**).
>
> Therefore, **extreme low-bit settings do not contradict our motivation; instead, they further highlight why our proposed components (preserving crucial information for Q’s calibration) are necessary**. Actually, we have also attempted the experiments below 3w3a before submission, and found that the optimization failed to converge in most cases, since it requires additional stabilization techniques beyond the scope of our method, which could serve as our future research focus.
>
> ---
>
> ### We greatly appreciate the reviewer’s time and insightful comments, and we hope our responses help clarify the concerns.

---

> > ### Author Rebuttal · Reviewer_dhFR · 2026-04-03
> >
> > Thank you for the detailed response. Most of my concerns are resolved. MaskAQ demonstrates performance gain compared with baseline MimiQ while taking a little more time and resources. I will keep my score.

---

> > > ### Author Response · Authors · 2026-04-04
> > >
> > > Thanks for your careful consideration and acknowledging our rebuttal. We mildly clarify that our MaskAQ introduces a modest and feasible computational overhead, while achieving clear performance gains, especially under ultra-low-bit settings. We again appreciate your time and thoughtful comments.

---

### Official Review · Reviewer_gJ99 · 2026-03-13

**Soundness:** 3
**Presentation:** 3
**Significance:** 3
**Originality:** 2
**Overall Recommendation:** 4
**Confidence:** 4

**Summary:**

The paper examines data-free quantization (DFQ) for Vision Transformers. It quantizes a pretrained full-precision model without using the original training data by creating calibration samples. The authors note that current DFQ methods often generate synthetic samples whose distributions do not match the inputs expected by the quantized model. This mismatch can result in lower performance. To address this issue, the authors introduce MaskAQ, a framework that targets patches with high attention values, which they call informative regions. The method uses an entropy-based regularization to differentiate between informative patches and background regions. It also incorporates a masked attention alignment loss that matches attention maps from the full-precision model to the quantized model for selected patches. The authors evaluate the method on ImageNet using various Vision Transformer architectures and multiple quantization settings. The results show improved accuracy compared to several existing DFQ methods across different bit-width configurations.

**Compliance With Llm Reviewing Policy:**

Affirmed.

**Key Questions For Authors:**

The paper proposes coupling selected informative regions with varying Q values to guide synthetic sample generation. Could the authors provide more intuition or ablation results on how the choice of masked regions affects quantization performance?

**Limitations:**

The authors could address potential limitations of MaskAQ, such as sensitivity to hyperparameters, scalability to very large ViTs, and/or performance on out-of-distribution data.

**Strengths And Weaknesses:**

Strengths:
+ The paper studies data-free quantization (DFQ) for Vision Transformers (ViTs) which is an important problem in scenarios where original training data cannot be accessed due to privacy or licensing constraints.
+ The proposed approach is motivated by the observation that attention in ViTs tends to concentrate on a subset of patches. The method focuses calibration and alignment on these "informative regions" using entropy-based regularization and masked attention alignment. The proposed design is reasonable (conceptually) and aligns with the structure of transformer attention mechanisms.
+ The overall method consists of understandable components (e.g. informative region decoupling, masked attention alignment, ...) that could be integrated into existing DFQ pipelines.
+ Experiments are conducted on several ViTs architectures and multiple quantization bit-widths. The paper also includes ablation studies examining the contributions of individual components. The appendix additionally reports experiments on object detection and semantic segmentation tasks which suggests that the approach may generalize beyond classification.

Weaknesses:
- Many elements of the method are related to ideas explored in prior DFQ work (e.g., patch similarity modeling, attention alignment, ..). The main difference is in masking the alignment to selected informative regions and introducing entropy-based regularization, which is an interesting idea but appears to be an incremental extension rather than a new approach.
- The reported results depend on assumptions about distributional similarity between real and synthetic samples that are difficult to verify in practice and are not empirically validated.
- Several aspects of the method rely on heuristic decisions (e.g. the selection of informative regions and masking thresholds). The paper provides a limited analysis on how sensitive the results are to these design choices.
- The paper motivation is to improve synthetic calibration samples, however, the evaluation mainly relies on qualitative visualizations. More quantitative analysis on how the generated samples match real data distributions would strengthen the claims.
- Minor: The theoretical part is somewhat disconnected from the practical algorithm.

---

> ### Author Rebuttal · Authors · 2026-03-31
>
> We sincerely thank Reviewer gJ99 for the constructive comments. Below are how we address your concerns and provide further clarification.
>
> ---
>
> ### **Q1**:  Many elements of the method are related to ideas explored in prior DFQ work (e.g., patch similarity modeling, attention alignment, ..).
>
> **A1**: We appreciate your comment. We kindly clarify the followings:
> - **Our innovation** is to tackle DFQ for ViT from **a novel information bottleneck perspective** (see **theoretical analysis in Sec.2.4**): *the mutual information dominates the primary principle of sample synthesis to crucially associate informative regions of synthetic samples with quantized model Q* (**Line 250-274**), rather than merely improving global realism or generic attention consistency in prior DFQ work.
> - This motivates us to resolve **the challenge on how the synthetic samples for ViTs preserve crucial information for Q’s calibration** (**Line 99-102**). To achieve it, the entropy term decouples informative regions from noisy background, while the masked alignment selectively couples these regions with Q, as clearly indicated in **Sec. 2.2** and **2.3**.
> - Therefore, MaskAQ is better viewed as **a novel calibration-oriented, region-selective, and Q-adaptive framework**, rather than an incremental extension.
>
> ---
>
> ### **Q2**:  The reported results depend on assumptions about distributional similarity between real and synthetic samples that are difficult to verify in practice and are not empirically validated.
> ### **Q3**:  The paper motivation is to improve synthetic calibration samples, however, the evaluation mainly relies on qualitative visualizations. More quantitative analysis on how the generated samples match real data distributions would strengthen the claims.
>
> **A2** & **A3**: Thanks for this comment. We mildly clarify the followings:
> - The results of MaskAQ **do not** depend on assumptions about distributional similarity between real and synthetic samples, since MaskAQ **shifts the goal of sample synthesis from approximating real data (prior DFQ work) to preserving the crucial information needed for calibrating Q**, as stated in **A1**.
> - Therefore, MaskAQ further focuses on the informative-region semantics of synthetic samples and their associated attention structure with Q's calibration, as clearly indicated in **Line 178-190**.
> - The quantitative results (see **Table 1 and 4**) across multiple backbones, bit-widths (especially for low bits), and downstream tasks exhibit the significant improvement of Q, which provide **indirect evidence** that **MaskAQ preserves calibration-relevant information for Q more effectively than prior DFQ work**.
>
> ---
>
> ### **Q4**:  Several aspects of the method rely on heuristic decisions. More intuition or ablation results on how the choice of masked regions affects quantization performance.
>
> **A4**: Thanks for the valuable suggestions. Actually, the masked regions serve as a strict subset of informative regions to prevent over-regularization (**Line 203-205**), where the mask ratio k (dynamically reduced from 50% to 10%) and the dropout probability ($p_{drop}=0.3$) are the main hyper-parameters  (**Line 321-324**). As you suggested, we conducted the sensitivity analsis with DeiT-B backbone under 3-bit cases below:
> | Mask ratio (k) | Acc. (%) |
> |--------|------|
> | 50%  | 41.95 |
> | 10%  | 42.53 |
> | 50%-10%  | **43.28**  |
>
> | $p_{drop}$ | 0.1 | 0.2 | 0.3 | 0.4 | 0.5 |
> |--------|------|------|------|------|------|
> | Acc. (%)  | 43.04 | 43.13 | **43.28** | 43.18  | 42.93 |
>
> The results imply that the dynamic schedule for k consistently *outperforms* fixed masking ratio and the masked regions   are *robust* to the choice of $p_{drop}$, which are consistent to the intuition of our adaptive mask.
>
> ---
>
> ### **Q5**:  Minor: The theoretical part is somewhat disconnected from the practical algorithm.
>
> **A5**:  Thanks for your comments, We kindly clarify that the theory is **not** intended as a disconnected add-on; it already serves as **the design basis for our two core losses**. Specifically, as **stated in section 2.4**, the **theoretical conditions are linked to the practical objectives** (Line 275-283): *first, $L_{fb}$ promotes semantic aggregation towards informative regions (Definition 1) within synthetic samples, capitalizing on the inherent sparsity of the self-attention mechanism, which is designed to satisfy the condition $|I(a_s;y)-I(a_r;y)|\le \xi$ in Eq.(22). Second, anchored on masked informative regions, $L_{align}$ enforces consistency between P and Q, helping satisfy the conditions in Eq.(20) and Eq.(21). Therefore, we can obtain desirable synthetic samples via Eq.(13) to benefit Q’s calibration*. As you suggested, we will refine the presentation of the theory-to-algorithm correspondence in revised manuscript.
>
>
> ---
>
> ### We thank the reviewer again for their valuable feedback and careful consideration, and we hope our clarifications address the raised concerns.

---

> > ### Author Rebuttal · Reviewer_gJ99 · 2026-04-04
> >
> > The authors have satisfactorily addressed my main concerns, especially through clearer explanations of the method and improvements in presentation. I appreciate the added detail and the effort made to resolve earlier ambiguities.

---

> > > ### Author Response · Authors · 2026-04-04
> > >
> > > Thanks for acknowledging our rebuttal. We are greatly honored that our clarifications and responses have addressed all your main concerns.  Many thanks for your appreciation to our contribution to the community!

---

### Decision · Program_Chairs · 2026-04-30

**Decision:**

Accept (regular)

**Comment:**

Based on the reviews and the authors’ rebuttal, I recommend **Accept** with low priority. The paper addresses a significant challenge in data-free quantization for Vision Transformers by shifting the synthesis objective from approximating real data to preserving calibration-relevant information via an information bottleneck perspective, leading to a novel masked attention alignment approach. The method demonstrates consistent and substantial performance gains across multiple backbones, bit-widths, and downstream tasks, with up to 3.1% Top-1 improvement on ImageNet. While one reviewer raised concerns about incremental novelty and theoretical justification, two reviewers explicitly confirmed that their concerns were fully resolved and maintained positive scores, and the rebuttal provided extensive additional experiments (ablation on larger models, computational cost analysis, sensitivity studies, and comparison of entropy surrogates) that address the key technical questions. The overall technical soundness, empirical rigor, and practical impact warrant acceptance.